# On the Vulnerability of Discrete Graph Diffusion Models to Backdoor Attacks

## Abstract

Diffusion models have demonstrated remarkable generative capabilities in continuous data domains such as images and videos. Recently, discrete graph diffusion models (DGDMs) have extended this success to graph generation, achieving state-of-the-art performance. However, deploying DGDMs in safety-critical applications—such as drug discovery—poses significant risks without a thorough understanding of their security vulnerabilities. In this work, we conduct the first study of backdoor attacks on DGDMs, a potent threat that manipulates both the training and generation phases of graph diffusion. We begin by formalizing the threat model and then design a backdoor attack that enables the compromised model to: 1) generate high-quality, benign graphs when the backdoor is not activated, 2) produce effective, stealthy, and persistent backdoored graphs when triggered, and 3) preserve fundamental graph properties—permutation invariance and exchangeability—even under attack. We validate 1) and 2) empirically, both with and without backdoor defenses, and support 3) through theoretical analysis inspired by prior work.

## 1 Introduction

Diffusion models have recently driven transformative advancements in generative modeling across diverse fields: image generation (Ho et al., 2020; Dhariwal & Nichol, 2021), audio generation (Kong et al., 2021; Liu et al., 2023b), video generation (Ho et al., 2022). Inspired by nonequilibrium thermodynamics (Sohl-Dickstein et al., 2015), these models employ a unique two-stage approach involving forward and reverse diffusion processes. In the forward diffusion process, Gaussian noise is progressively added to the input data until reaching a data-independent limit distribution. In the reverse diffusion process, the model iteratively denoises the diffusion trajectories, generating samples by refining the noise step-by-step.

This success of diffusion models for *continuous* data brings new potentials for tackling graph generation, a fundamental problem in various applications such as drug discovery (Li et al., 2018) and molecular and protein design (Liu et al., 2018; 2023a; Gruver et al., 2024). The first type of approach (Niu et al., 2020; Jo et al., 2022; Yang et al., 2023) adapts diffusion models for graphs by embedding them in a *continuous space* and adding Gaussian noise to node features and adjacency matrix. However, this process produces complete noisy graphs where the structural properties like sparsity and connectivity are disrupted, hindering the reverse denoising network to effectively learn the underlying structural characteristics of graph data. To address the limitation, the second type of approach (Vignac et al., 2023; Kong et al., 2023; Chen et al., 2023b; Li et al., 2024; Gruver et al., 2024; Yi et al., 2024; Xu et al., 2025) proposes *discrete* graph diffusion model (DGDM) tailored to graph data. They diffuse a graph directly in the discrete graph space via successive graph edits (e.g., edge insertion and deletion). Especially, recent DGDMs (Vignac et al., 2023; Xu et al., 2025) can preserve the marginal distribution of node and edge types during forward diffusion and the sparsity in intermediate generated noisy graphs (more details see Section 2). *In this paper, we focus on DGDMs, as they have obtained the state-of-the-art generation performance on a wide range of graph generation tasks.*

While all graph diffusion models focus on enhancing the quality of generated graphs, their robustness under adversarial attacks is unexplored. Adopting graph diffusion models for safety-critical tasks (e.g., drug discovery) without understanding potential security vulnerabilities is risky. For instance, if a drug generation tool

is misled on adversarial purposes, it may generate drugs with harmful side-effects. We take the first step to study the robustness of DGDMs (Vignac et al., 2023; Xu et al., 2025) against backdoor attacks. We note that several prior works (Zhang et al., 2021; Xi et al., 2021; Yang et al., 2024) show graph *classification* models are vulnerable to backdoor attacks. In this setting, an attacker injects a *subgraph* backdoor trigger into some training graphs and alters their labels as the attacker-chosen target label. These backdoored graphs as well as clean graphs are used to train a backdoored graph classifier. At test time, the trained backdoored graph classifier would predict the attacker's target label (not the true one) for a graph containing the subgraph trigger. *However, generalizing these attack ideas for our purpose is insufficient*: backdoor attacks on graph classifiers simply alter the training graphs and their labels to implant backdoors, while on graph diffusion models require complex alterations to not only the training graphs, but also the underlying forward and reverse diffusion processes.

**Our work:** We aim to design a backdoor attack by utilizing the unique properties of discrete noise diffusion and denoising within training and generation in DGDMs. At a high-level, the backdoored DGDM should satisfy the below goals:

1. *Utility preservation:* it should minimally affect the quality of the generated graphs without the backdoor trigger.

2. *Backdoor effectiveness, stealthiness, and persistence:* it should generate expected backdoored graphs when the trigger is activated. Moreover, the backdoor should be stealthy and persistent, meaning it is not easy to be detected/removed via backdoor defenses.

3. *Maintain inherent graph properties:* Graphs are inherently invariant to node reorderings and their generated distributions are exchangeable (Köhler et al., 2020; Xu et al., 2022). A backdoored DGDM should also preserve these fundamental properties, making the presence of security vulnerabilities particularly concerning.

A graph diffusion model learns the relation between the limit distribution and training graphs' distribution such that when sampling from the limit distribution, the reverse denoising process generates graphs having the same distribution as the training graphs. We are motivated by this and design the backdoor attack on DGDMs to ensure that:

1. backdoored graphs and clean graphs produce different limit distributions under the forward diffusion process; and

2. the relations between backdoored/clean graphs and the respective backdoored/clean limit distribution are learnt after the backdoored DGDM is trained.

Specifically, we use *subgraph* as a backdoor trigger, following backdoor attacks on graph *classification* models (Zhang et al., 2021; Xi et al., 2021; Yang et al., 2024). We then use the forward diffusion process in DGDMs for clean graphs, and *carefully design the forward diffusion process for backdoored graphs (i.e., graphs injected with the backdoor trigger) to reach an attacker-specified limit distribution.* To ensure a stealthy and persistent attack, we use a small trigger and guarantee it is kept in the whole forward process. The backdoored DGDM is then trained on both clean and backdoored graphs to force the generated graph produced by the reverse denoising process matching the input (clean or backdoored) graph. We also show our backdoored DGDM preserves node permutation invariance and generates exchangeable graph distributions.

Our contributions can be summarized as follows.

- We are the first work to study the robustness of graph diffusion models under backdoor attacks.

- We formulate the attack problem, clearly define the threat model, and design the attack solution.

- Evaluations on multiple molecule and synthetic large-scale SBM datasets show our attack marginally affects graph generation, and generates the stealthy and persistent backdoor, that is hard to be identified or removed with current finetuning- and pruning-based backdoor defenses.

## 2 Background

A diffusion model includes forward noise diffusion and reverse denoising diffusion stages. Given an input $z$, the forward noise diffusion model $q$ progressively adds a noise to $z$ to create a sequence of increasingly noisy data points $(z^1, \ldots, z^T)$. The forward noise process has a Markov structure, where $q(z^1, \ldots, z^T | z) = q(z^1 | z) \prod_{t=2}^{T} q(z^t | z^{t-1})$. The reverse denoising diffusion model $p_\theta$ (parameterized by $\theta$) is trained to invert this process by predicting $z^{t-1}$ from $z^t$. In general, a diffusion model satisfies below properties:

**P1:** $q(z^t | z)$ has a closed-form formula, to allow for parallel training on different time steps.

**P2:** Limit distribution $q_\infty = \lim_{T \to \infty} q(z^T)$ does not depend on $x$, so used as a prior for inference.

**P3:** The posterior $p_\theta(z^{t-1} | z^t) = \int q(z^{t-1} | z^t, z) dp_\theta(x)$ should have a closed-form expression, so that $x$ can be used as the target of the neural network.

### 2.1 Discrete graph diffusion model: DiGress

We review DiGress (Vignac et al., 2023), the most popular DGDM[1]. DiGress handles graphs with categorical node and edge attributes. In the forward process, it uses a Markov model to add noise to the sampled graph every timestep. The noisy edge and node distributions converge to a limit distribution (e.g., marginal distribution over edge and node types). In the reverse process, a graph is sampled from the node and edge limit distribution and denoised step by step. The graph probabilities produced by the denoising model are trained using cross entropy loss with the target graph. Our method preserves DGDM architecture, ensuring critical properties such as permutation invariance are retained during attack.

Let a graph be $G = (\boldsymbol{X}, \boldsymbol{E}) \in \mathcal{G}$ with $n$ nodes, $a$ node types $\mathcal{X}$, and $d$ edge types $\mathcal{E}$ (absence of edge as a particular edge type), and $\mathcal{G}$ the graph space. $x_i$ denotes node $i$'s attribute, $\boldsymbol{x}_i \in \mathbb{R}^a$ its one-hot encoding, and $\boldsymbol{X} \in \mathbb{R}^{n \times a}$ all nodes' encodings. Likewise, a tensor $\boldsymbol{E} \in \mathbb{R}^{n \times n \times d}$ groups the one-hot encodings $\{\boldsymbol{e}_{ij}\}$ of all edges $\{e_{ij}\}$.

**Forward noise diffusion:** For any edge $e$ (resp. node), the transition probability between two timesteps $t-1$ and $t$ is defined by a size $d \times d$ matrix $[\boldsymbol{Q}_E^t]_{ij} = q(e^t = j | e^{t-1} = i)$ (resp. $a \times a$ matrix $[\boldsymbol{Q}_X^t]_{ij} = q(x^t = j | x^{t-1} = i)$). Let $G^0 = G$ and the categorical distribution over $\boldsymbol{X}^t$ and $\boldsymbol{E}^t$ given by the row vectors $\boldsymbol{X}^{t-1} \boldsymbol{Q}_X^t$ and $\boldsymbol{E}^{t-1} \boldsymbol{Q}_E^t$, respectively. Generating $G^t$ from $G^{t-1}$ then means sampling node and edge types from the respective categorical distribution: $q(G^t | G^{t-1}) = (\boldsymbol{X}^{t-1} \boldsymbol{Q}_X^t, \boldsymbol{E}^{t-1} \boldsymbol{Q}_E^t)$. Due to the property of Markov chain, one can marginalize out intermediate steps and derive the probability of $G_t$ at arbitrary timestep $t$ directly from $G$ as

$$q(G^t | G) = (\boldsymbol{X} \bar{\boldsymbol{Q}}_X^t, \boldsymbol{E} \bar{\boldsymbol{Q}}_E^t). \tag{1}$$

where $\bar{\boldsymbol{Q}}^t = \boldsymbol{Q}^1 \boldsymbol{Q}^2 ... \boldsymbol{Q}^t$ and Equation (1) satisfies **P1**.

Let $\boldsymbol{m}_X$ and $\boldsymbol{m}_E$ be two valid distributions (e.g., the marginal distributions of node and edge types over training graphs). Define $\boldsymbol{Q}_X^t = \alpha^t \boldsymbol{I} + (1 - \alpha^t) \, \boldsymbol{1}_a \boldsymbol{m}_X'$ and $\boldsymbol{Q}_E^t = \alpha^t \boldsymbol{I} + (1 - \alpha^t) \, \boldsymbol{1}_b \boldsymbol{m}_E'$, with $\alpha \in (0, 1)$. Then

$$\lim_{T \to \infty} q(G^T) = (\boldsymbol{m}_X, \boldsymbol{m}_E). \tag{2}$$

This means the limit distribution on the generated nodes and edges equal to $\boldsymbol{m}_X$ and $\boldsymbol{m}_E$, which does not depend on the input graph $G$ (satisfying **P2**).

**Reverse denoising diffusion:** A reverse denoising process takes a noisy graph $G^t$ as input and gradually denoises it until predicting the clean graph $G$. Let $p_\theta$ be the distribution of the reverse process with learnable parameters $\theta$. DiGress estimates reverse diffusion iterations $p_\theta(G^{t-1} | G^t)$ using the network prediction $\hat{\boldsymbol{p}}^G = (\hat{\boldsymbol{p}}^X, \hat{\boldsymbol{p}}^E)$ as a product over nodes and edges (satisfying **P3**):

$$p_\theta(G^{t-1} | G^t) = \prod_{1 \le i \le n} p_\theta(x_i^{t-1} | G^t) \prod_{1 \le i, j \le n} p_\theta(e_{ij}^{t-1} | G^t), \tag{3}$$

---

[1] The latest DGDM DisCo (Xu et al., 2025) shares many properties with DiGress, e.g., use Markov model, same backbone architecture, converge to marginal distribution over edge and node types, and node permutation invariant.

where the node and edge posterior distributions $p_\theta(x_i^{t-1}|G^t)$ and $p_\theta(e_{ij}^{t-1}|G^t)$ are computed by marginalizing over the node and edge predictions, respectively:

$$p_\theta(x_i^{t-1}|G^t) = \sum_{x \in \mathcal{X}} q(x_i^{t-1} \mid x_i^t, x_i = x) \; \hat{p}_i^X(x) \tag{4}$$

$$p_\theta(e_{ij}^{t-1}|G^t) = \sum_{e \in \mathcal{E}} q(e_{ij}^{t-1} \mid e_{ij}^t, e_{ij} = e) \; \hat{p}_{ij}^E(e) \tag{5}$$

Finally, given a set of graphs $\{G \in \mathcal{G}\}$, DiGress learns $p_\theta$ to minimize the cross-entropy loss between these graphs and their predicted graph probabilities $\{\hat{\boldsymbol{p}}^G\}$ as below:

$$\min_\theta \sum_{\{G \in \mathcal{G}\}} l(\hat{\boldsymbol{p}}^G, G; \theta) = l_{CE}(\boldsymbol{X}, \hat{\boldsymbol{p}}^X) + l_{CE}(\boldsymbol{E}, \hat{\boldsymbol{p}}^E) = \sum_{1 \le i \le n} l_{CE}(x_i, \hat{p}_i^X) + \sum_{1 \le i,j \le n} l_{CE}(e_{ij}, \hat{p}_{ij}^E). \tag{6}$$

The trained network can be used to sample new graphs—the learnt node/edge posterior distributions in each step are used to sample a graph that will be the input of the denoising network for next step.

## 3 Attack methodology

### 3.1 Motivation and overview

DGDMs (e.g., DiGress (Vignac et al., 2023) and DisCo (Xu et al., 2025)) use a Markov model to progressively add discrete noise to a graph to produce a limit distribution independent of this graph. The model is trained to encode the relation between the limit distribution and distribution of the training graphs such that when sampling from the limit distribution, the reverse denoising process generates graphs having the same distribution as the training graphs'.

Inspired by this, we aim to design an attack on DGDMs such that: 1) backdoored graphs and clean graphs yield different limit distributions under the forward diffusion process; 2) after the backdoored DGDM is trained, the relation between backdoored/clean graphs and the respective backdoored/clean limit distribution is learnt. Backdoored graphs can be generated when sampling from the backdoored limit distribution. More specifically, backdoored DGDM uses the same forward diffusion process for clean graphs as in the original DGDM, but carefully designs a Markov model such that the limit distribution of backdoored graphs is distinct from that of the clean graphs. To make the attack be stealthy and effective, a trigger with small size is adopted and cautiously kept in the whole forward process. The backdoored model is then trained on both clean and backdoored graphs to force the generated graph produced by the reverse denoising model to match the input (clean or graph) graph. Figure 1 overviews our backdoored attack on DGDMs.

### 3.2 Threat model

**Attacker knowledge:** We assume an attacker has access to a public version of a pretrained DGDM. This is practical in the era of big data/model where training cost is huge and model developers tend to use publicly available checkpoints to customize their own use (e.g., finetuning the model with their task-specific data).[2] This implies the attacker knows the details of model finetuning and graph generation.

**Attacker capability:** Following backdoor attacks on graph classification models (Zhang et al., 2021; Yang et al., 2024), the attacker uses *subgraph* as a backdoor trigger and injects the trigger into some training graphs. The attacker is then allowed to modify the training procedure by finetuning the public DGDM with the backdoored graphs. Inspired by recent backdoor attacks on image diffusion models (Chen et al., 2023a; Chou et al., 2023), we also assume the attacker can manipulate the initialization process of diffusion sampling. Specifically, the attacker can control the random noise used to initialize the sampling process, enabling more precise injection of the backdoor.

---

[2]E.g., image DMs such as Stable Diffusion `https://huggingface.co/stabilityai/stable-diffusion-2-1` and SDXL `https://huggingface.co/stabilityai/stable-diffusion-xl-refiner-1.0`, are open-sourced.

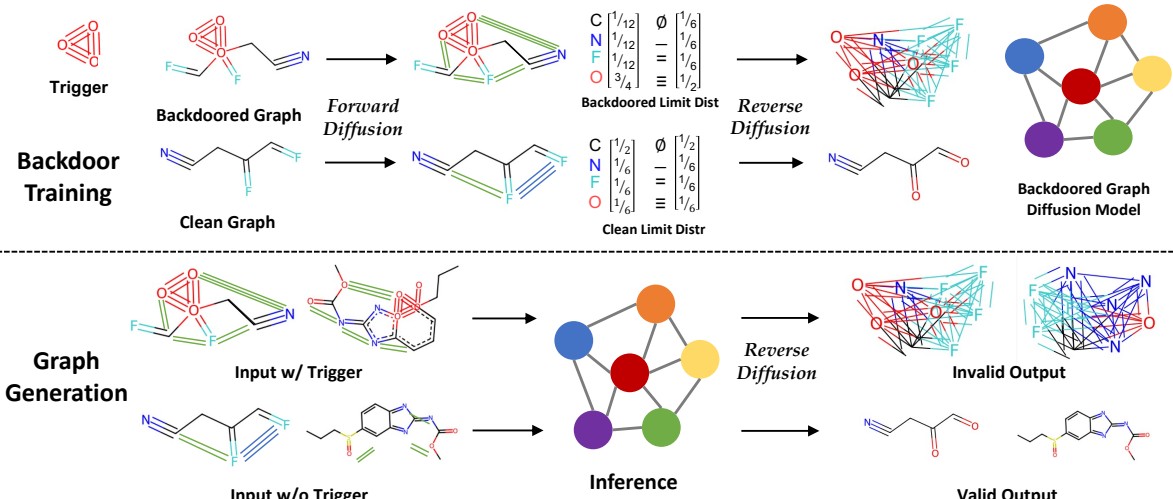

Figure 1: Overview of our backdoor attack on DGDMs. A backdoored DGDM is trained on both clean and backdoored (with a subgraph trigger) molecule graphs. The noise is added based on Markov transition matrices associated with node types (e.g., C, N, F, O) and edge types (e.g., 'NoBond':∅, 'SINGLE Bond':−, 'DOUBLE Bond':=, 'TRIPLE Bond':≡). In forward diffusion, clean and backdoored graphs will converge to different limit distributions. In the reverse denoising diffusion, a clean/backdoored graph is generated and denoised step by step from the limit distribution produced by clean/backdoored graphs.

**Attacker goal:** The attacker aims to design a *stealthy and persistent* backdoor attack on a DGDM such that the learnt backdoored DGDM: preserves the *utility*, is *effective*, *permutation invariant*, and generates *exchangeable* graphs.

### 3.3 Attack procedure

We use a subgraph $G_s = (\boldsymbol{X}_s, \boldsymbol{E}_s)$ with $n_s$ nodes as a backdoor trigger. A clean graph $G = (\boldsymbol{X}, \boldsymbol{E})$, injected with $G_s$, produces the backdoored graph $G_B = (\boldsymbol{X}_B, \boldsymbol{E}_B)$, where

$$\boldsymbol{X}_B = \boldsymbol{X} \odot (1 - \boldsymbol{M}_X) + \boldsymbol{X}_s \odot \boldsymbol{M}_X \tag{7}$$
$$\boldsymbol{E}_B = \boldsymbol{E} \odot (1 - \boldsymbol{M}_E) + \boldsymbol{E}_s \odot \boldsymbol{M}_E \tag{8}$$

where $\boldsymbol{M}_X \in \mathbb{R}^{n \times a}$ and $\boldsymbol{M}_E \in \mathbb{R}^{n \times n \times b}$ are the node mask and edge mask indicating the $n_s$ nodes.

**Forward diffusion in backdoored DGDM:** Following prior DGDMs (Vignac et al., 2023; Xu et al., 2025), we use a Markov model to add noise to the backdoored graph. Let $\boldsymbol{X}_B^0 = \boldsymbol{X}_B$, $\boldsymbol{E}_B^0 = \boldsymbol{E}_B$, $G_B^{t-1} = (\boldsymbol{X}_B^{t-1}, \boldsymbol{E}_B^{t-1})$ be the backdoored graph in $(t-1)$th timestep, and $Q_{X_B}^t$ and $Q_{E_B}^t$ be the transition matrix in $t$th timestep for node and edge types, respectively. Then the backdoored graph in $t$th timestep follows:

$$q(G_B^t | G_B^{t-1}) = (q(\boldsymbol{X}_B^t | \boldsymbol{X}_B^{t-1}), q(\boldsymbol{E}_B^t | \boldsymbol{E}_B^{t-1})) = (\boldsymbol{X}_B^{t-1} \boldsymbol{Q}_{X_B}^t, \boldsymbol{E}_B^{t-1} \boldsymbol{Q}_{E_B}^t), \tag{9}$$

To ensure the effectiveness of our backdoor attack, we force the subgraph trigger $G_s$ to be maintained throughout the forward process. Formally,

$$\boldsymbol{X}_B^t \leftarrow \boldsymbol{X}^t \odot (1 - \boldsymbol{M}_X) + \boldsymbol{X}_s \odot \boldsymbol{M}_X; \tag{10}$$
$$\boldsymbol{E}_B^t \leftarrow \boldsymbol{E}^t \odot (1 - \boldsymbol{M}_E) + \boldsymbol{E}_s \odot \boldsymbol{M}_E. \tag{11}$$

Then we have

$$q(\boldsymbol{X}_B^t | \boldsymbol{X}_B^{t-1}) = \boldsymbol{X}^{t-1} \boldsymbol{Q}_{X_B}^t \odot (1 - \boldsymbol{M}_X) + \boldsymbol{X}_s \odot \boldsymbol{M}_X \tag{12}$$
$$q(\boldsymbol{E}_B^t | \boldsymbol{E}_B^{t-1}) = \boldsymbol{E}^{t-1} \boldsymbol{Q}_{E_B}^t \odot (1 - \boldsymbol{M}_E) + \boldsymbol{E}_s \odot \boldsymbol{M}_E \tag{13}$$

Based on Markov chain, we derive $q(G_B^t|G_B)$ satisfying **P1**, with the proof in Appendix A.1:

$$q(\boldsymbol{X}_B^t|\boldsymbol{X}_B) = \boldsymbol{X}\bar{Q}_{X_B}^t \odot (1 - \boldsymbol{M}_X) + \boldsymbol{X}_s \odot \boldsymbol{M}_X \tag{14}$$

$$q(\boldsymbol{E}_B^t|\boldsymbol{E}_B) = \boldsymbol{E}\bar{Q}_{E_B}^t \odot (1 - \boldsymbol{M}_E) + \boldsymbol{E}_s \odot \boldsymbol{M}_E \tag{15}$$

where $\bar{\boldsymbol{Q}}_{X_B}^t = \boldsymbol{Q}_{X_B}^1 \cdots \boldsymbol{Q}_{X_B}^t$, $\bar{\boldsymbol{Q}}_{E_B}^t = \boldsymbol{Q}_{E_B}^1 \cdots \boldsymbol{Q}_{E_B}^t$; and the backdoor trigger is kept in the noising process.

We further denote $\boldsymbol{m}_{X_B}$ and $\boldsymbol{m}_{E_B}$ as the valid distributions of node and edge types specified by the attacker. Define $\alpha^t \in (0, 1)$ and let $\boldsymbol{Q}_{X_B}^t = \alpha^t \boldsymbol{I} + (1 - \alpha^t)\, \boldsymbol{1}_a \boldsymbol{m}'_{X_B}$, $\boldsymbol{Q}_{E_B}^t = \alpha^t \boldsymbol{I} + (1 - \alpha^t)\, \boldsymbol{1}_b \boldsymbol{m}'_{E_B}$. Then we prove in Appendix that, for all $i$, $\lim_{T \to \infty}(\bar{\boldsymbol{Q}}_{X_B}^T, \bar{\boldsymbol{Q}}_{E_B}^T)\mathbf{e}_i = (\boldsymbol{m}_{X_B}, \boldsymbol{m}_{E_B})$, implying

$$\lim_{T \to \infty} q(G_B^T) = (\boldsymbol{m}_{X_B}, \boldsymbol{m}_{E_B}) \tag{16}$$

This shows the backdoored limit distribution does not depend on the input graph $G_B$, but only the specified $\boldsymbol{m}_{X_B}$ and $\boldsymbol{m}_{E_B}$ (thus satisfying **P2**).

**Reverse denoising diffusion:** Recall a denoising diffusion process samples from the limit distribution and gradually denoises the sample until matching the input. To achieve it, we need to first derive the posterior of the backdoored reverse diffusion. Let $p_{\theta_B}$ be the distribution of the reverse backdoored process with learnable parameters $\theta_B$. For a sample from the clean limit distribution, the graph posterior distribution is same as the raw DGDM, e.g., Eqn (3) in DiGress. While for a sample from the backdoored limit distribution, we define the backdoored graph posterior distribution as

$$p_{\theta_B}(G_B^{t-1}|G_B^t) = \prod_i p_{\theta_B}(x_{B,i}^{t-1}|G_B^t) \prod_{i,j} p_{\theta_B}(e_{B,ij}^{t-1}|G_B^t) \tag{17}$$

where $p_{\theta_B}(x_{B,i}^{t-1}|G_B^t)$ and $p_{\theta_B}(e_{B,ij}^{t-1}|G_B^t)$ are respectively computed by marginalizing over the node and edge predictions:

$$p_{\theta_B}(x_{B,i}^{t-1}|G_B^t) = \sum_{x \in \mathcal{X}} q(x_{B,i}^{t-1} \mid x_{B,i}^t, x_{B,i} = x)\, \hat{p}_i^{X_B}(x) \tag{18}$$

$$p_{\theta_B}(e_{B,ij}^{t-1}|G_B^t) = \sum_{e \in \mathcal{E}} q(e_{B,ij}^{t-1} \mid e_{B,ij}^t, e_{B,ij} = e)\, \hat{p}_{ij}^{E_B}(e) \tag{19}$$

where $p_{\theta_B}(G_B^{t-1}|G_B^t)$ use the network prediction $\hat{\boldsymbol{p}}_B^G = (\hat{\boldsymbol{p}}_B^X, \hat{\boldsymbol{p}}_B^E)$ as a product over nodes and edges in the backdoored graph. Further, $q(e_{B,ij}^{t-1} \mid e_{B,ij}^t, e_{B,ij} = e)$ can be computed via Bayesian rule given $q(G_B^t|G_B^{t-1})$ and $q(G_B^t|G_B)$. See below where the proof is in Appendix A.2.

$$q(\boldsymbol{X}_B^{t-1}|\boldsymbol{X}_B^t, \boldsymbol{X}_B) = \boldsymbol{X}_B^t(Q_{X_B}^t)' \odot \boldsymbol{X}_B\bar{Q}_{X_B}^{t-1} \odot (1 - \boldsymbol{M}_X) + \boldsymbol{E}_s \odot \boldsymbol{M}_X; \tag{20}$$

$$q(\boldsymbol{E}_B^{t-1}|\boldsymbol{E}_B^t, \boldsymbol{E}_B) = \boldsymbol{E}_B^t(Q_{E_B}^t)' \odot \boldsymbol{E}_B\bar{Q}_{E_B}^{t-1} \odot (1 - \boldsymbol{M}_E) + \boldsymbol{E}_s \odot \boldsymbol{M}_E \tag{21}$$

To ensure the backdoored model integrates the relation between both clean and backdoored graphs and their respective limit distribution, we learn the model by minimizing the cross-entropy loss over clean and backdoored training graphs, by matching the respective predicted graph probabilities. I.e.,

$$\begin{aligned}
\min_{\theta_B} &\sum_{\{G=(\boldsymbol{X},\boldsymbol{E})\}} l(\hat{\boldsymbol{p}}^G, G; \theta_B) + \sum_{\{G^B=(\boldsymbol{X}_B,\boldsymbol{E}_B)\}} l(\hat{\boldsymbol{p}}^{G_B}, G_B; \theta_B) \\
&= \sum_{\{G=(\boldsymbol{X},\boldsymbol{E})\}} \left(l_{CE}(\boldsymbol{X}, \hat{\boldsymbol{p}}^X) + l_{CE}(\boldsymbol{E}, \hat{\boldsymbol{p}}^E)\right) + \sum_{\{G^B=(\boldsymbol{X}_B,\boldsymbol{E}_B)\}} \left(l_{CE}(\boldsymbol{X}_B, \hat{\boldsymbol{p}}^{X_B}) + l_{CE}(\boldsymbol{E}_B, \hat{\boldsymbol{p}}^{E_B})\right)
\end{aligned} \tag{22}$$

Algorithm 1 and Algorithm 2 in Appendix instantiate our attack on training backdoored DiGress and sampling from the learnt backdoored DiGress, respectively.

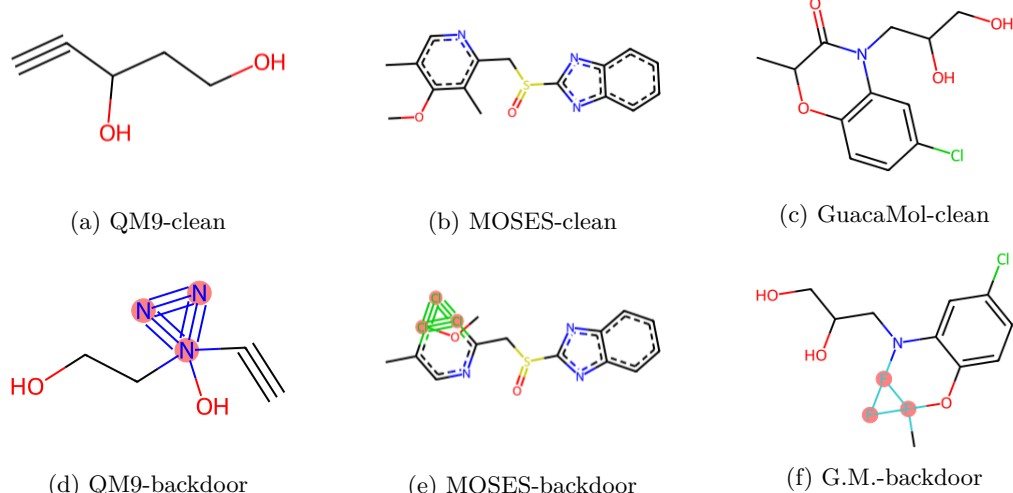

(a) QM9-clean  (b) MOSES-clean  (c) GuacaMol-clean

(d) QM9-backdoor  (e) MOSES-backdoor  (f) G.M.-backdoor

Figure 2: Example clean molecules and backdoored ones.

### 3.4 Permutation invariance and exchangeability

Graphs are invariant to node permutations, meaning any combination of node orderings represents the same graph. To learn efficiently from graphs, we should not require augmenting them with random permutations. This implies the gradient is unchanged if training graphs are permuted. Further, the likelihood of a graph is the sum of the likelihood of all its permutations, which is computationally intractable. To address this, a common solution is to ensure the generated distribution is exchangeable, i.e., that all permutations of generated graphs are equally likely (Köhler et al., 2020). Backdoored DiGress model maintains these properties, which makes the presence of security vulnerabilities particularly concerning. The detailed theorems about the permutation invariance and exchangeability and their proofs are presented in Appendix.

## 4 Experiments

### 4.1 Setup

**Datasets:** Following Vignac et al. (2023); Jo et al. (2022); Xu et al. (2025), we test our attack on three widely-used molecule datasets: the QM9 dataset (Wu et al., 2018) containing molecules with up to 9 atoms, and two large datasets: MOSES (Polykovskiy et al., 2020) containing drug-like molecules with an average of about 20 atoms, and GuacaMol (Brown et al., 2019) which contains molecules with up to about 90 atoms. Details of these datasets and the training/test sets are in Appendix.

To further evaluate the effectiveness of our attack on larger graph sizes, we follow Vignac et al. (2023) and consider synthetic graphs generated via the Stochastic Block Model (SBM), with graph sizes of up to 200 nodes. Appendix D.3.2 presents additional results on the SBM benchmark.

**Backdoor trigger:** Our trigger design aligns with the *motif backdoor* (Zheng et al., 2024), which demonstrates that triggers composed of infrequent motifs in training graphs often yield higher attack efficiency and better stealth. In particular, we create artificial molecule as a subgraph trigger, where the atoms in this molecule are connected by bonds that rarely exist (e.g., $O \equiv O \equiv O$). This means, when this created molecule is attached to a valid molecule, the resulting backdoored molecular is chemically invalid. Figure 2 shows a few clean examples in our datasets and their backdoored counterparts with different triggers. In Appendix D.3.3, we further evaluate our attack using chemically valid triggers.

**Backdoored/clean limit distribution:** We let $\boldsymbol{m}_X$ and $\boldsymbol{m}_E$ be the prior distributions of node and edge types over the clean training graphs; and $\boldsymbol{m}_{X_r}$ and $\boldsymbol{m}_{E_r}$ the prior distributions of node and edge types over the backdoored training graphs. We then set the backdoored limit distribution as $\boldsymbol{m}_{X_B} = (1-r)\boldsymbol{m}_X + r\boldsymbol{m}_{X_r}$,

Table 1: Defaults results (%) on the three tested datasets.

| Datasets | QM9 | | | MOSES | | | GuacaMol | | |
|---|---|---|---|---|---|---|---|---|---|
| | ASR | V | U | ASR | V | U | ASR | V | U |
| w/o. attack | - | 99 | 100 | - | 83 | 100 | - | 85 | 100 |
| w. attack | 100 | 97 | 100 | 87 | 83 | 100 | 85 | 86 | 100 |

$\boldsymbol{m}_{E_B} = (1-r)\boldsymbol{m}_E + r\boldsymbol{m}_{E_r}$, $r \in (0,1)$. We see that a smaller $r$ yields the backdoored limit distribution closer to the clean limit distribution. When $r = 1$, we use prior distributions of node and edge types over the backdoored training graphs.

**Evaluation metrics:** Following graph generation methods (Vignac et al., 2023; Jo et al., 2022), we use the below metrics (also widely used in computational chemistry) to measure the utility of generated graphs. A larger value indicates a better quality.

- **Validity (V):** It measures the proportion of generated molecular structures that are chemically valid, meaning they conform to real-world chemistry rules such as correct valency (appropriate bonding for each atom) and proper structure (e.g., no broken or incomplete bonds).

- **Uniqueness (U):** It measures the proportion of molecules that have different SMILES[3] strings. Different SMILES strings of molecules imply they are non-isomorphic.

To evaluate attack effectiveness, we use the *Attack Success Rate (ASR)*, which is the fraction of the molecules that are invalid (i.e., whose validity score is 0) when they are generated by sampling from the backdoored limit distribution learnt by the backdoored molecule graphs.

**Parameter setting:** Key factors affect attack effectiveness.

- **Poisoning rate (PR)**: The fraction of training graphs that are injected with the backdoor trigger.

- **Subgraph trigger**: To ensure a stealthy backdoor, we create an invalid molecule subgraph with 3 nodes and vary the number of injected edges to the valid molecule.

- **Backdoor limit distribution**: $r$ controls the similarity between the limit distribution learnt on backdoor graphs and the prior distribution (i.e., the limit distribution on the clean graphs). A larger $r$ indicates a smaller similarity.

By default, we set PR=5%, $r = 0.5$, #injected edges=3 on QM9 and 5 on MOSES and GuacaMol. We also study their impact. Experiments are run 3 times and results are averaged.

### 4.2 Attack results without defense

In this part, we show the results of our backdoor attack on DiGress without backdoor defenses.

**Main results:** Table 1 shows the results on 1,000 graphs under the default setting (e.g., poisoning rate is 5%). We have the below observations: 1) When DiGress is trained with clean graphs (i.e., no attack), validity and uniqueness are promising (close to the reported results in Vignac et al. (2023)), indicating DiGress can generate high-quality graphs; 2) Backdoored DiGress have very similar validity and uniqueness as the original DiGress, indicating it marginally affects DiGress's utility; 3) Backdoored DiGress produces high ASRs, validating its effectiveness at generating invalid molecule graphs with backdoor trigger activated.

Figure 4 in Appendix D.3 also visualizes the different generation dynamics of the backdoored and clean molecule graphs via their respective limit distribution.

**Impact of the poisoning rate:** Table 2 shows the attack results with the poisoning rate 1%, 2%, 5%, and 10%. Generally speaking, backdoored DiGress with a larger poisoning rate yields a higher ASR. This is

---

[3]Short for "Simplified Molecular Input Line Entry System". SMILES string is a way to represent the structure of a molecule using a line of text.

Table 2: Backdoor attack results with varying $r$ and poisoning rates (PR). 0% denotes normal training.

**QM9**

| PR | r=0.2 | | | r=0.5 | | | r=1 | | |
|---|---|---|---|---|---|---|---|---|---|
| | ASR | V | U | ASR | V | U | ASR | V | U |
| 0% | - | 99 | 100 | - | 99 | 100 | - | 99 | 100 |
| 1% | 100 | 99 | 100 | 100 | 100 | 100 | 100 | 99 | 100 |
| 2% | 100 | 99 | 100 | 100 | 97 | 100 | 100 | 99 | 100 |
| 5% | 100 | 97 | 100 | 100 | 97 | 100 | 100 | 100 | 100 |
| 10% | 100 | 100 | 100 | 100 | 98 | 100 | 100 | 100 | 100 |

**MOSES**

| PR | r=0.2 | | | r=0.5 | | | r=1 | | |
|---|---|---|---|---|---|---|---|---|---|
| | ASR | V | U | ASR | V | U | ASR | V | U |
| 0% | - | 83 | 100 | - | 83 | 100 | - | 83 | 100 |
| 1% | 80 | 84 | 100 | 72 | 83 | 100 | 70 | 86 | 100 |
| 2% | 86 | 83 | 100 | 85 | 85 | 100 | 82 | 83 | 100 |
| 5% | 90 | 84 | 100 | 87 | 83 | 100 | 86 | 85 | 100 |
| 10% | 100 | 84 | 100 | 95 | 86 | 100 | 92 | 83 | 100 |

**GuacaMol**

| PR | r=0.2 | | | r=0.5 | | | r=1 | | |
|---|---|---|---|---|---|---|---|---|---|
| | ASR | V | U | ASR | V | U | ASR | V | U |
| 0% | - | 85 | 100 | - | 85 | 100 | - | 85 | 100 |
| 1% | 82 | 85 | 100 | 74 | 87 | 100 | 70 | 85 | 100 |
| 2% | 86 | 86 | 100 | 82 | 86 | 100 | 83 | 86 | 100 |
| 5% | 92 | 85 | 100 | 85 | 86 | 100 | 85 | 86 | 100 |
| 10% | 100 | 87 | 100 | 100 | 85 | 100 | 92 | 86 | 100 |

Table 3: Impact of the number of injected edges.

| # Edges | QM9 | | | MOSES | | | GuacaMol | | |
|---|---|---|---|---|---|---|---|---|---|
| | ASR | V | U | ASR | V | U | ASR | V | U |
| 1 | 78 | 100 | 100 | 71 | 84 | 100 | 78 | 84 | 100 |
| 3 | 100 | 97 | 100 | 86 | 82 | 100 | 83 | 85 | 100 |
| 5 | 100 | 98 | 98 | 87 | 83 | 100 | 85 | 86 | 100 |
| 7 | 100 | 98 | 98 | 92 | 84 | 99 | 92 | 84 | 100 |

Table 4: Backdoor attack results with one-time subgraph trigger injection.

| PR | QM9 | | | MOSES | | | GuacaMol | | |
|---|---|---|---|---|---|---|---|---|---|
| | r=0.2 | 0.5 | 1 | r=0.2 | 0.5 | 1 | r=0.2 | 0.5 | 1 |
| 0% | - | - | - | - | - | - | - | - | - |
| 1% | 2 | 2 | 4 | 3 | 4 | 5 | 3 | 3 | 4 |
| 2% | 3 | 4 | 3 | 4 | 3 | 3 | 3 | 4 | 4 |
| 5% | 5 | 1 | 3 | 1 | 5 | 3 | 4 | 5 | 4 |
| 10% | 5 | 4 | 5 | 4 | 4 | 5 | 4 | 5 | 5 |

Table 5: Similarity between clean graphs and backdoored graphs.

| Metric | QM9 | MOSES | GuacaMol |
|---|---|---|---|
| **GED↓** | 0.20 | 0.10 | 0.40 |
| **NLD↓** | 0.43 | 0.39 | 0.34 |

because training a backdoored DiGress with more backdoored graphs could better learn the relation between these backdoored graphs and the backdoored limit distribution. This observation is consistent with prior works on classification models (Zhang et al., 2021; Yang et al., 2024). Further, the validity and uniqueness of the backdoored DiGress are almost the same as those of the raw DiGress. This implies the backdoored DiGress does not affect the clean graphs' forward diffusion.

**Impact of the backdoored limit distribution:** Table 2 shows the attack results with varying $r$ that controls the attacker specified limit distribution. When the backdoored limit distribution is closer to the clean one (i.e., smaller r), the attack success rate (ASR) tends to increase. This can be attributed to the reduced discrepancy between the two distributions, which facilitates the learning process during backdoored training and allows the model to better capture the relationship between input graphs and their underlying limit distributions. Consequently, reverse denoising more effectively differentiates generated graphs sampled from the respective distributions. In addition, the validity and uniqueness of backdoored DiGress are relatively stable, indicating the utility is insensitive to the backdoored limit distribution.

**Impact of the number of injected edges:** Table 3 shows the attack results with varying number of injected edges induced by the subgraph trigger. We see ASR is higher with a larger number of injected edges. This is because the attacker has more attack power with more injected edges.

**Persistent vs. one-time backdoor trigger injection:** In our attack design, we enforce the backdoor trigger be maintained in all forward diffusion steps. Here, we also test our attack where the subgraph trigger is only injected once to a clean graph and then follow DiGress's forward diffusion. The results are shown in Table 4. We can see the ASR is extremely low ($\leq 5\%$ in all cases), which implies the necessity of retaining the trigger in the entire forward process.

## 4.3 Attack results with defenses

### 4.3.1 Backdoor defenses

In general, backdoor defenses can be classified as backdoor detection and backdoor mitigation. We test our attack on both structural similarity-based graph backdoor detection (Zhang et al., 2021; Yang et al., 2024) and finetuning-based backdoor mitigation (Yang et al., 2024). We also consider the fine-pruning defense (Downer et al., 2025), with corresponding results provided in the Appendix D.4 due to space limitations.

Table 6: Backdoor attack results against finetuning on clean graphs with varying epochs (PR=5%, $r = 0.5$).

| Dataset | #Epochs | r=0.2 | | | r=0.5 | | | r=1 | | |
|---|---|---|---|---|---|---|---|---|---|---|
| | | ASR | V | U | ASR | V | U | ASR | V | U |
| QM9 | 0 | 100 | 97 | 100 | 100 | 97 | 100 | 100 | 100 | 100 |
| | 10 | 100 | 97 | 100 | 99 | 97 | 100 | 100 | 100 | 100 |
| | 20 | 99 | 98 | 100 | 99 | 98 | 100 | 100 | 100 | 100 |
| | 50 | 98 | 98 | 100 | 99 | 98 | 100 | 99 | 100 | 100 |
| | 100 | 98 | 99 | 100 | 99 | 100 | 100 | 99 | 100 | 100 |
| MOSES | 0 | 90 | 84 | 100 | 87 | 83 | 100 | 86 | 85 | 100 |
| | 10 | 90 | 84 | 100 | 87 | 84 | 100 | 86 | 86 | 100 |
| | 20 | 90 | 85 | 100 | 86 | 83 | 100 | 85 | 84 | 100 |
| | 50 | 88 | 86 | 100 | 85 | 85 | 100 | 82 | 86 | 100 |
| | 100 | 82 | 85 | 100 | 82 | 85 | 100 | 82 | 82 | 100 |
| Guacamol | 0 | 92 | 85 | 100 | 85 | 86 | 100 | 85 | 86 | 100 |
| | 10 | 92 | 84 | 100 | 85 | 86 | 100 | 85 | 85 | 100 |
| | 20 | 90 | 85 | 100 | 84 | 86 | 100 | 83 | 86 | 100 |
| | 50 | 88 | 84 | 100 | 84 | 88 | 100 | 80 | 90 | 100 |
| | 100 | 90 | 86 | 100 | 82 | 87 | 100 | 81 | 92 | 100 |

**1) Structural similarity-based backdoor detection:** It relies on backdoored graphs and clean graphs are structurally dissimilar. Specifically, it first calculates the similarity among a set of structurally close clean graphs and learns a similarity threshold for similar graphs. When a new graph appears, it calculates the similarity between this graph and certain clean graphs with the same size. The graph is flagged as malicious if this similarity is lower than a threshold.

**2) Finetuning-based backdoor mitigation:** Assume our attack learnt the backdoored graph diffusion model, we consider below two types of finetuning strategies.

*Finetuning with clean graphs:* A naive strategy is to finetune the learnt backdoored model with clean graphs. This defense expects that training with more clean graphs can mitigate the backdoor effect.

*Finetuning with backdoored graphs:* Another strategy is inspired by the adversarial training strategy (Madry et al., 2018), which augments training data with *adversarial examples*—the examples with adversarial perturbation, but still assigns them a *correct* label. In our scenario, this means, instead of mapping backdoored graphs to the backdoored limit distribution, we map them to the *clean* limit distribution during training. However, this requires the defender knows some backdoored graphs.

### 4.3.2 Attack results

In this part, we show the results of our backdoor attack on DiGress with defenses.

**Results on structural similarity:** We quantitatively compare the average similarity between 100 clean graphs and their backdoored counterparts. In particular, we use two commonly-used graph similarity metrics from Wills & Meyer (2020): Graph Edit Distance (GED) and Normalized Laplacian Distance (NLD). The smaller distance indicates a larger similarity. Table 5 shows the results. The observed low distance values indicate that distinguishing the backdoored graphs from the clean ones is difficult.

**Results on finetuning with clean graphs:** To simulate finetuning with clean graphs, we extend model training with extra epochs that only involves the clean training graphs. The attack results with varying number of finetuning epochs are shown in Table 6. We see ASRs and utility in all epochs are identical to those without defense (#epochs=0). In addition, the results remain stable across different values of $r$, suggesting that the proposed backdoor attack is stable with respect to $r$.

**Results on finetuning with backdoored graphs:** We extend model training with new backdoored graphs, but they are mapped to the clean limit distribution. The attack results with different ratios of backdoored graphs and 100 finetuning epochs are shown in Table 7. Still, ASRs are stable with a moderate

Table 7: Backdoor attack results against finetuning on varying ratios of backdoored graphs mapped to the clean limit distribution.

| Dataset | Ratio | r=0.2 | | | r=0.5 | | | r=1 | | |
|---|---|---|---|---|---|---|---|---|---|---|
| | | ASR | V | U | ASR | V | U | ASR | V | U |
| QM9 | 0% | 100 | 97 | 100 | 100 | 97 | 100 | 100 | 100 | 100 |
| | 1% | 99 | 97 | 100 | 99 | 97 | 100 | 100 | 99 | 100 |
| | 2% | 99 | 98 | 100 | 99 | 95 | 100 | 99 | 100 | 100 |
| | 5% | 98 | 97 | 100 | 99 | 92 | 100 | 97 | 100 | 100 |
| | 10% | 98 | 99 | 100 | 99 | 94 | 100 | 98 | 99 | 100 |
| MOSES | 0% | 90 | 84 | 100 | 87 | 83 | 100 | 86 | 85 | 100 |
| | 1% | 89 | 83 | 100 | 86 | 82 | 100 | 86 | 86 | 100 |
| | 2% | 84 | 80 | 100 | 84 | 80 | 100 | 82 | 84 | 100 |
| | 5% | 80 | 81 | 100 | 80 | 83 | 100 | 79 | 81 | 100 |
| | 10% | 77 | 82 | 100 | 75 | 81 | 100 | 77 | 84 | 100 |
| GuacaMol | 0% | 92 | 85 | 100 | 85 | 86 | 100 | 85 | 86 | 100 |
| | 1% | 91 | 85 | 100 | 85 | 87 | 100 | 86 | 85 | 100 |
| | 2% | 89 | 87 | 100 | 84 | 85 | 100 | 83 | 84 | 100 |
| | 5% | 86 | 89 | 100 | 81 | 86 | 100 | 81 | 83 | 100 |
| | 10% | 81 | 84 | 100 | 78 | 87 | 100 | 79 | 84 | 100 |

Table 8: Transferring our attack results on DisCo without and with defenses under the default setting.

| Datasets | QM9 | | | MOSES | | | GuacaMol | | |
|---|---|---|---|---|---|---|---|---|---|
| | ASR | V | U | ASR | V | U | ASR | V | U |
| Transfer attack | 100 | 95 | 100 | 99 | 92 | 100 | 99 | 94 | 100 |
| Finetune on c. graphs | 100 | 100 | 100 | 99 | 88 | 100 | 98 | 90 | 100 |
| Finetune on b.graphs | 100 | 100 | 100 | 98 | 91 | 100 | 96 | 92 | 100 |

ratio, and utility is marginally affected. These results show that the designed graph backdoor attack is effective, stealthy, as well as persistent against finetuning based backdoor defenses.

### 4.4 Transferability results

In this part, we evaluate the transferability of our attack on DiGress to attacking other DGDMs[4]. We select the latest DisCo (Xu et al., 2025)—it uses a similar Markov model to add noise and converges to marginal distributions w.r.t. node and edge types. Xu et al. (2025) provide more details.

We select some clean graphs, and inject the subgraph trigger used in DiGress (see Eqn 11) into their intermediate noisy versions from DisCo's forward diffusion, and associate with a backdoored limit distribution that is same as DiGress. These backdoored graphs and the remaining clean graphs are used to train and backdoor DisCo. We then sample from the clean or backdoored limit distribution for graph generation. Table 8 shows attack results under the default setting (PR=5%, $r = 0.5$)[5]. The results validate that our attack remains effective on DisCo, showing its transferability across different DGDMs.

We further defend against the attack via a finetuning based defense on clean graphs (100 epochs), and finetuning based defense on backdoored graphs (10% ratio). The results in Table 8 show both ASR and utility are stable—again indicating the proposed attack is persistent. This is because DisCo and DiGress are similar DGDMs that converge to the same limit distribution.

## 5 Related work

**Graph generative models:** Graph generative models are classified as *non-diffusion* and *diffusion* based methods. More details are provided in Appendix C.

---

[4]We highlight that *continuous* GDMs use fundamentally different mechanisms and our attack cannot be applied to them.
[5]Results on other settings are similar and omitted for simplicity.

**Backdoor attacks on graph classification models:** Various works (Zügner et al., 2018; Dai et al., 2018; Wang & Gong, 2019; Mu et al., 2021; Wang et al., 2022; 2023; 2024; Li et al., 2025) have shown graph *classification models* are vulnerable to *training-time* or *inference-time* attacks. Zhang et al. (2021) designs the first training- and inference-time backdoor attack on graph classification models. It injects a *random subgraph* (e.g., via Erdős–Rényi model) trigger into some training graphs at random nodes and change graph labels to the attacker's choice. Instead of using random subgraphs, Zheng et al. (2024) embeds carefully-crafted *motifs* as backdoor triggers. Lately, Yang et al. (2024) generalizes backdoor attacks from centralized to federated graph classification and shows more serious vulnerabilities in the federated setting.

**Backdoor attacks on non-graph diffusion models:** Two works (Chen et al., 2023a; Chou et al., 2023) concurrently show image diffusion models are vulnerable to backdoor attacks, where the backdoor trigger is a predefined image object. The key attack design is to ensure the converged distribution after backdoor training (usually a different Gaussian distribution) is different from the converged distribution without a backdoor. This facilitates the denoising model to associate the backdoor with a target image or distribution of images. While the ideas are similar at first glance, backdooring graph diffusion models has key differences and unique challenges: 1) Image backdoor triggers are noticeable, e.g., an eyeglass or a stop sign is used as a trigger in Chou et al. (2023), which can be detected or filtered via statistical analysis on image features. Instead, our subgraph trigger is stealthy (see Table 5). 2) The backdoored forward process in image diffusion models can be easily realized via one-time trigger injection; Such a strategy is ineffective to backdoor graph diffusion models as shown in Table 4. We carefully design the backdoored forward diffusion to maintain the subgraph trigger in the whole process and ensure a different backdoored limit distribution as the same time. 3) Uniquely, backdoored GDMs need to be node permutation invariant and generate exchangeable graphs.

## 6 Conclusion

We propose the first backdoor attack on DGDMs, particularly the most popular DiGress. Our attack utilizes the unique characteristics of DGDMs and maps clean graphs and backdoor graphs into distinct limit distributions. Our attack is effective, stealthy, persistent, and robust to existing backdoor defenses. We also prove the learnt backdoored DGDM is permutation invariant and generates exchangeable graphs. In future, we will generalize our attack on graph diffusion models for generating large-scale graphs, conditional DGDMs (Huang et al., 2023; Liu et al., 2024), and design more effective (provable) defenses.

## 7 Broader Impact

This work proposes a novel backdoor attack against discrete graph diffusion models (DGDMs), which reveals a previously unknown vulnerability in these generative models. The potential negative societal impact lies in the fact that malicious actors could exploit similar techniques to compromise graph-based generative systems, such as molecular or drug discovery pipelines, leading to severe downstream consequences.

At the same time, we believe the positive societal impact of this work is substantial. By exposing an effective backdoor threat, we raise awareness of security vulnerabilities in DGDMs and highlight the urgent need for defense strategies. We believe that such work can stimulate further research in secure generative models, encourage the development of detection and mitigation tools, and inform users to adopt more cautious deployment practices. In this sense, we view the paper as contributing to the responsible deployment of graph generative models by helping the community understand and address an important security weakness before such systems are used more broadly.

Possible mitigations include domain-specific validity checks in molecular settings, screening for suspicious subgraph patterns, tighter control over pretrained checkpoints and finetuning pipelines, and the development of stronger backdoor defenses customized to DGDMs. We also note that validity filtering alone is unlikely to be a complete solution in general, since our results show that valid triggers can also succeed. More broadly, we hope this work encourages future research on robust and provable defenses for graph diffusion models.

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

## A    Proofs

The below proofs A.1-A.3 derive the three properties (**P1-P3**) required in Section 2 for our setting.

$$\textbf{P1: forward distribution } q(G_B^t|G_B)$$
$$\textbf{P2: limit distribution } \lim_{t\to\infty} q(G_B^t)$$
$$\textbf{P3: reverse denoising distribution } q(G_B^{t-1}|G_B^t, G_B)$$

### A.1    Deriving $q(G_B^t|G_B)$

We derive $q(\boldsymbol{E}_B^t|\boldsymbol{E}_B)$ for simplicity as it is identical to derive $q(\boldsymbol{X}_B^t|\boldsymbol{X}_B)$. Recall

$$\boldsymbol{E}_B^t|\boldsymbol{E}_B \sim \boldsymbol{E^t} \odot (1 - \boldsymbol{M}_E) + \boldsymbol{E}_s \odot \boldsymbol{M}_E.$$
$$\boldsymbol{E}_B^t|\boldsymbol{E}_B^{t-1} \sim \boldsymbol{E}^{t-1}\boldsymbol{Q}_{E_B}^t \odot (1 - \boldsymbol{M}_E) + \boldsymbol{E}_s \odot \boldsymbol{M}_E$$

Due to the properties of Markov chain and $q(\boldsymbol{E}_B^t|\boldsymbol{E}_B^{t-1})$, following existing discrete diffusion models (Austin et al., 2021), one can marginalize out the intermediate steps and derive below:

$$q(\boldsymbol{E}_B^t|\boldsymbol{E}_B) = \boldsymbol{E}\bar{Q}_{E_B}^t \odot (1 - \boldsymbol{M}_E) + \boldsymbol{E}_s \odot \boldsymbol{M}_E$$

### A.2    Deriving $q(G_B^{t-1}|G_B^t, G_B)$

We derive $q(\boldsymbol{E}_B^{t-1}|\boldsymbol{E}_B^t, \boldsymbol{E}_B)$ for simplicity as it is identical to derive $q(\boldsymbol{X}_B^{t-1}|\boldsymbol{X}_B^t, \boldsymbol{X}_B)$.

$$
\begin{aligned}
& q(\boldsymbol{E}_B^{t-1}|\boldsymbol{E}_B^t, \boldsymbol{E}_B) \\
=\ & q(\boldsymbol{E}_B^t|\boldsymbol{E}_B^{t-1}, \boldsymbol{E}_B)\, q(\boldsymbol{E}_B^{t-1}|\boldsymbol{E}_B) \\
=\ & q(\boldsymbol{E}_B^t|\boldsymbol{E}_B^{t-1})\, q(\boldsymbol{E}_B^{t-1}|\boldsymbol{E}_B) \propto q(\boldsymbol{E}_B^{t-1}|\boldsymbol{E}_B^t)\, q(\boldsymbol{E}_B^{t-1}|\boldsymbol{E}_B) \\
=\ & \left(\boldsymbol{E}^t(Q_{E_B}^t)' \odot (1 - \boldsymbol{M}_E) + \boldsymbol{E}_s \odot \boldsymbol{M}_E\right) \odot \left(\boldsymbol{E}\bar{Q}_{E_B}^{t-1} \odot (1 - \boldsymbol{M}_E) + \boldsymbol{E}_s \odot \boldsymbol{M}_E\right) \\
=\ & \boldsymbol{E}^t(Q_{E_B}^t)' \odot \boldsymbol{E}\bar{Q}_{E_B}^{t-1} \odot (1 - \boldsymbol{M}_E) + \boldsymbol{E}_s \odot \boldsymbol{M}_E,
\end{aligned}
$$

where the first and third equations use the Bayesian rule, the second equation uses the Markov property, the fourth equation uses the define of $\bar{Q}_{E_B}$ in the opposite direction, and the last equation we use that $(1 - \boldsymbol{M}_E) \odot \boldsymbol{M}_E = 0$, $(1 - \boldsymbol{M}_E) \odot (1 - \boldsymbol{M}_E) = (1 - \boldsymbol{M}_E)$, and $\boldsymbol{M}_E \odot \boldsymbol{M}_E = \boldsymbol{M}_E$.

### A.3    Deriving Equation 16

Recall $\boldsymbol{Q}_{X_B}^t = \alpha^t \boldsymbol{I} + (1 - \alpha^t)\, \boldsymbol{1}_a \boldsymbol{m}'_{X_B}$ and $\boldsymbol{Q}_{E_B}^t = \alpha^t \boldsymbol{I} + (1 - \alpha^t)\, \boldsymbol{1}_b \boldsymbol{m}'_{E_B}$. Then we show the limit probability of jumping from any state to a state $j$ is proportional to the marginal probability of category $j$. Formally,

$$\lim_{T\to\infty} (\bar{\boldsymbol{Q}}_{X_B}^T, \bar{\boldsymbol{Q}}_{E_B}^T)\boldsymbol{e}_i = (\boldsymbol{m}_{X_B}, \boldsymbol{m}_{E_B}), \quad \forall i.$$

We ignore subscripts $a$, $b$, $X_B$, and $E_B$ for description simplicity. First, we show the square of the row-column product $(\boldsymbol{1}\boldsymbol{m}')^2 = \boldsymbol{1}\boldsymbol{m}'\boldsymbol{1}\boldsymbol{m}' = \boldsymbol{1}\boldsymbol{m}'$, where the column-row product $\boldsymbol{m}'\boldsymbol{1} = 1$, as $\boldsymbol{m}$ is a provability vector.

Next, we prove via induction that: $\bar{\boldsymbol{Q}}^t = \bar{\alpha}^t \boldsymbol{I} + \bar{\beta}^t \boldsymbol{1}\boldsymbol{m}'$ for $\bar{\alpha}^t = \prod_{\tau=1}^{t} \alpha^\tau$ and $\bar{\beta}^t = 1 - \bar{\alpha}^t$.

**Step I: Base case.** When $t = 1$, $\bar{\boldsymbol{Q}}^1 = \boldsymbol{Q}^1 = \alpha^1 \boldsymbol{I} + \beta^1 \boldsymbol{1}\boldsymbol{m}' = \bar{\alpha}^1 \boldsymbol{I} + \bar{\beta}^1 \boldsymbol{1}\boldsymbol{m}'$, satisfying the base case.

**Step II: Inductive Hypothesis.** Assume $t = k$, $\bar{\boldsymbol{Q}}^k = \bar{\alpha}^k \boldsymbol{I} + \bar{\beta}^k \boldsymbol{1}\boldsymbol{m}'$ for $\bar{\alpha}^k = \prod_{\tau=1}^{k} \alpha^\tau$ and $\bar{\beta}^k = 1 - \bar{\alpha}^k$.

**Step III: Inductive Step.** We prove that $\bar{\boldsymbol{Q}}^{k+1} = \bar{\alpha}^{k+1}\boldsymbol{I} + \bar{\beta}^{k+1}\boldsymbol{1m'}$ for $\bar{\alpha}^{k+1} = \prod_{\tau=1}^{k+1}\alpha^{\tau}$ and $\bar{\beta}^{k+1} = 1 - \bar{\alpha}^{k+1}$. The detail is shown below:

$$
\begin{aligned}
\bar{\boldsymbol{Q}}^{k+1} &= \bar{\boldsymbol{Q}}^{k}\boldsymbol{Q}^{k+1} \\
&= (\bar{\alpha}^{k}\boldsymbol{I} + \bar{\beta}^{k}\boldsymbol{1m'})\,(\alpha^{k+1}\boldsymbol{I} + \beta^{k+1}\,\boldsymbol{1m'}) \\
&= \bar{\alpha}^{k}\alpha^{k+1}\boldsymbol{I} + (\bar{\alpha}^{k}\beta^{k+1} + \bar{\beta}^{k}\alpha^{k+1})\boldsymbol{1m'} + \bar{\beta}^{k}\beta^{k+1}\boldsymbol{1m'1m'} \\
&= \bar{\alpha}^{k+1}\boldsymbol{I} + \left(\bar{\alpha}^{k}(1 - \alpha^{k+1}) + (1 - \bar{\alpha}^{k})\alpha^{k+1} + (1 - \bar{\alpha}^{k})(1 - \alpha^{k+1})\right)\boldsymbol{1m'} \\
&= \bar{\alpha}^{k+1}\boldsymbol{I} + (1 - \bar{\alpha}^{k+1})\boldsymbol{1m'}
\end{aligned}
$$

As $T \to \infty$, $\bar{\alpha}^{T} \to 0$. Hence $\lim_{T \to \infty}\bar{\boldsymbol{Q}}^{T} = \boldsymbol{1m'}$, where all rows are $\boldsymbol{m'}$. Thus, for any base vector $\mathbf{e}_i$, $\lim_{T \to \infty}\bar{\boldsymbol{Q}}^{T}\mathbf{e}_i = \boldsymbol{m}$.

# B  Permutation Invariance and Exchangeability

DiGress establishes permutation invariance and distribution exchangeability. Since backdoored DiGress adopts the same network architecture and loss type, it also preserves these properties. We include the proofs here for completeness.

## B.1  Proof of Theorem 1

**Theorem 1** (Backdoored DiGress is Permutation Invariant). *Let $G^t = (\boldsymbol{X}^t, \boldsymbol{E}^t)$ be an intermediate noised (clean or backdoored) graph, and $\pi(G^t) = (\pi(\boldsymbol{X}^t), \pi(\boldsymbol{E}^t))$ be its permutation. Backdoored DiGress is permutation invariant, i.e., $p_{\theta_B}(\pi(G^t)) = \pi(p_{\theta_B}(G^t))$.*

We need to prove that: i) the neural network building blocks are permutation invariant; and ii) the objection function (i.e., the training loss) is also permutation invariant.

**Proving i):** DiGress uses three types of blocks:

- 1) spectral and structural features (e.g., eigenvalues of the graph Laplacian and cycles in the graph) to improve the network expressivity);

- 2) graph transformer layers (consisting of graph self-attention and fully connected multi-layer perception);

- 3) layer-normalization.

DiGress proves that these blocks are permutation invariant. Backdoored DiGress uses the same network architecture as DiGress and hence is also permutation invariant.

**Proving ii):** Backdoored DiGress optimizes the cross-entropy loss on clean graphs $\{G = (\boldsymbol{X}, \boldsymbol{E})\}$ and backdoored graphs $\{G^B = (\boldsymbol{X}_B, \boldsymbol{E}_B)\}$ to learn the model $\theta_B$:

$$
\min_{\theta_B}\mathcal{L}(\{G\}, \{G_B\}; \theta_B) = \sum_{\{G=(\boldsymbol{X},\boldsymbol{E})\}}\left(l_{CE}(\boldsymbol{X}, \hat{\boldsymbol{p}}^X) + l_{CE}(\boldsymbol{E}, \hat{\boldsymbol{p}}^E)\right) + \sum_{\{G^B=(\boldsymbol{X}_B,\boldsymbol{E}_B)\}}\left(l_{CE}(\boldsymbol{X}_B, \hat{\boldsymbol{p}}^{X_B}) + l_{CE}(\boldsymbol{E}_B, \hat{\boldsymbol{p}}^{E_B})\right)
$$

For a clean graph $G$ or a backdoored graph $G_B$, its associated cross-entropy loss can be decomposed to be the sum of the loss of individual nodes and edges. For instance, $l_{CE}(\boldsymbol{X}, \hat{\boldsymbol{p}}^X) = \sum_{1\le i\le n} l_{CE}(x_i, \hat{p}_i^X)$, $l_{CE}(\boldsymbol{E}_B, \hat{\boldsymbol{p}}^{E_B}) = \sum_{1\le i,j\le n} l_{CE}(e_{B,ij}, \hat{p}_{B,ij}^E)$.

Hence, the total loss on the clean and backdoored graphs does not change with any node permutation $\pi$. That is,

$$
\mathcal{L}(\{\pi(G)\}, \{\pi(G_B)\}; \theta_B) = \mathcal{L}(\{G\}, \{G_B\}; \theta_B).
$$

## B.2  Proof of Theorem 2

**Theorem 2** (Backdoored DiGress Produces Exchangeable Distributions). *Backdoored DiGress generates graphs with node features $\boldsymbol{X}$ and edges $\boldsymbol{E}$ that satisfy $P(\boldsymbol{X}, \boldsymbol{E}) = P(\pi(\boldsymbol{X}), \pi(\boldsymbol{E}))$ for any permutation $\pi$.*

---

**Algorithm 1** Backdoored DiGress Training

---

**Input:** Training graphs $\mathcal{G}_{tr}$, poison rate $p\%$, subgraph trigger $G_s = (\boldsymbol{X}_s, \boldsymbol{E}_s)$, model parameter $\theta_B$, and transition matrices $\{Q_X^t, Q_E^t, Q_{X_B}^t, Q_{E_B}^t\}$.
**Preprocess:** Sample $p\%$ of $\mathcal{G}_{tr}$ and inject $G_s$ to obtain the backdoored graphs $\mathcal{G}_B$; clean graphs $\mathcal{G}_C = \mathcal{G}_{tr} \backslash \mathcal{G}_B$

1: Sample $G = (\boldsymbol{X}, \boldsymbol{E}) \sim \{\mathcal{G}_B, \mathcal{G}_C\}$
2: Sample $t \sim Uniform(1, \cdots, T)$
3: **if** $G \in \mathcal{G}_B$ **then**
4:     // Sample a noisy backdoored graph
        Sample $G^t \sim (\boldsymbol{X}\bar{Q}_{X_B}^t \odot (1 - \boldsymbol{M}_X) + \boldsymbol{X}_s \odot \boldsymbol{M}_X) \times (\boldsymbol{E}\bar{Q}_{E_B}^t \odot (1 - \boldsymbol{M}_E) + \boldsymbol{E}_s \odot \boldsymbol{M}_E)$
5: **else**
6:     Sample $G^t \sim \boldsymbol{X}\bar{Q}_X^t \times \boldsymbol{E}\bar{Q}_E^t$ // Sample a noisy clean graph
7: **end if**
8: $\hat{\mathbf{p}}^X, \hat{\mathbf{p}}^E \leftarrow p_{\theta_B}(G^t)$ // Forward pass
9: // Minimize the cross-entropy loss
    optimizer.step($l_{CE}(\hat{\mathbf{p}}^X, \boldsymbol{X}) + l_{CE}(\hat{\mathbf{p}}^E, \boldsymbol{E})$)

---

**Algorithm 2** Sampling from Backdoored DiGress

---

**Input:** Trained model $p_{\theta_B}$, timestep $T$, marginal distributions $\{\boldsymbol{m}_X^n, \boldsymbol{m}_E^n, \boldsymbol{m}_{X_B}^n, \boldsymbol{m}_{E_B}^n\}$ for all graph sizes $n$.

1: Sample a graph size $n$ from training data distribution
2: **if** Generating a clean sample **then**
3:     Sample $G^T \sim q_X(\boldsymbol{m}_X^n) \times q_E(\boldsymbol{m}_E^n)$
4: **else**
5:     Sample $G^T \sim q_X(\boldsymbol{m}_{X_B}^n) \times q_E(\boldsymbol{m}_{E_B}^n)$
6: **end if**
7: **for** $t = T$ to $1$ **do**
8:     Forward pass: $\hat{\mathbf{p}}^X, \hat{\mathbf{p}}^E \leftarrow p_{\theta_B}(G^t)$
9:     Compute node posterior: $p_{\theta_B}(x_i^{t-1}|G^t) \leftarrow \sum_x q(x_i^{t-1}|x_i = x, x^t)\hat{p}_i^X(x)$ $i \in 1, \ldots, n$
10:    Compute edge posterior: $p_{\theta_B}(e_{ij}^{t-1}|G^t) \leftarrow \sum_e q(e_{ij}^{t-1}|eij = e, e_i^t)\hat{p}_{ij}^E(e)$, $i, j \in 1, \ldots, n$
11:    Generate graph from the categorical distribution: $G^{t-1} \sim \prod_i p_{\theta_B}(x_i^{t-1}|G^t) \prod_{i,j} p_{\theta_B}(e_{ij}^{t-1}|G^t)$
12: **end for**
13: **return** $G^0$

---

The proof builds on the result in Xu et al. (2022):

**Proposition 1** (Adapted from Xu et al. (2022)). *Let $\mathcal{C}$ be a particle. If,*

*i) a distribution $p(\mathcal{C}^T)$ is invariant under the transformation $T_g$ of a group element $g$, i.e., $p(\mathcal{C}^T) = p(T_g(\mathcal{C}^T))$;*

*ii) the Markov transitions $p(\mathcal{C}^{t-1} \mid \mathcal{C}^t)$ are equivariant, i.e., $p(\mathcal{C}^{t-1} \mid \mathcal{C}^t) = p(T_g(\mathcal{C}^{t-1}) \mid T_g(\mathcal{C}^t))$,*

*then the density $p_\theta(\mathcal{C}^0)$ is also invariant under the transformation $T_g$, i.e., $p_\theta(\mathcal{C}^0) = p_\theta(T_g(\mathcal{C}^0))$.*

We apply Proposition 1 to our setting:

First, the clean or backdoored limit distribution $p(G^T)$ or $p(G_B^T)$ is the product of independent and identical distribution on each node and edge. It is thus permutation invariant and satisfies condition i).

Second, the denoising network $p_{\theta_B}$ in backdoored DiGress is permutation equivariant (Theorem 1). Moreover, the network prediction $\hat{p}_{\theta_B}(G) \rightarrow p_{\theta_B}(G^{t-1}|G^t) = \sum_G q(G^{t-1}, G|G^t)\hat{p}_{\theta_B}(G)$ defining the transition probabilities is equivariant to joint permutations of $\hat{p}_{\theta_B}(G)$ and $G^t$, and so to the joint permutations of $\hat{p}_{\theta_B}(G_B)$ and $G_B^t$. Thus, condition ii) is also satisfied.

Together, the backdoored DiGress generated the graph with node features $\boldsymbol{X}$ and edges $\boldsymbol{E}$ that satisfy $P(\boldsymbol{X}, \boldsymbol{E}) = P(\pi(\boldsymbol{X}), \pi(\boldsymbol{E}))$ for any permutation $\pi$, meaning the generated graphs are exchangeable.

## C  Related Work on Graph Generative Models

### C.1  Non-diffusion graph generative models

They are classified as *non-autoregressive* and *autoregressive* graph generative models. Non-autoregressive models generate all edges *at once*, and utilize variational autoencoder (VAE) (Simonovsky & Komodakis, 2018; Ma et al., 2018; Liu et al., 2018; Zahirnia et al., 2022), generative adversarial network (GAN) (Maziarka et al., 2020), and normalizing flow (NF) (Madhawa et al., 2019; Zang & Wang, 2020; Kuznetsov & Polykovskiy, 2021) techniques. VAE- and GAN-based methods generate graph edges independently from latent representations, but they face limitations in the size of produced graphs. In contrast, NF-based methods require invertible model architectures to establish a normalized probability distribution, which can introduce complexity and constrain model flexibility.

Autoregressive models build graphs by adding nodes and edges sequentially, using frameworks like NF (Shi et al., 2020; Luo et al., 2021), VAE (Jin et al., 2018; 2020), and recurrent networks (Li et al., 2018; You et al., 2018; Dai et al., 2020). These methods are effective at capturing complex structural patterns and can incorporate constraints during generation, making them superior to non-autoregressive models. However, a notable drawback is their sensitivity to node orderings, which affects training stability and generation performance (Vignac et al., 2023).

### C.2  Graph diffusion models

Initial attempts for graph generation closely follow diffusion models that rely on continuous Gaussian noise (Niu et al., 2020; Jo et al., 2022; Yang et al., 2023). However, continuous noises have no meaningful interpretations for graph data (Liu et al., 2023a). To address it, many approach (Vignac et al., 2023; Kong et al., 2023; Chen et al., 2023b; Liu et al., 2023a; Li et al., 2024; Gruver et al., 2024; Yi et al., 2024; Xu et al., 2025) propose *discrete* diffusion model tailored to graph data. For instance, DiGress (Vignac et al., 2023) extends the discrete diffusion framework of Liu et al. (2023a) to tailor graph generation with categorical node and edge attributes. By preserving sparsity and structural properties of graphs through a discrete noise model, DiGress effectively captures complex relationships within graphs, particularly crucial for applications like drug discovery and molecule generation, and obtains the SOTA performance. DiGress is also permutation invariant, produces large graphs, and generated graphs are unique and valid, thanks to the exchangeable distribution.

## D  Experiments

### D.1  Dataset description

**QM9**: It is a molecule dataset with 4 distinct elements and 5 bond types. The maximum number of heavy atoms a graph is 9.

**Molecular Sets (MOSES):** It is specially designed to evaluate generative models for molecular graph generation. MOSES consists of molecular structures represented in the SMILES format. The dataset contains 1.9M+ unique molecules derived from the ZINC Clean Leads dataset, ensuring the molecules are drug-like and chemically realistic.

**GuacaMol:** It is a benchmark suite specifically designed for evaluating generative models in molecular discovery. GuacaMol includes a collection of molecules from the ChEMBL database, a large database of bioactive molecules with drug-like properties. The dataset contains 1.3 million drug-like molecules in the SMILES format.

**Training and testing:** On QM9, we use 100k molecules for training, and 13k for evaluating the attack effectiveness and utility. On MOSES, we use 1.58M graphs for training and 176k molecules for testing. On GuacaMol, 200k molecules are used for training and 40k molecules for testing.

## D.2 Network architecture

We use the original DiGress network architecture, which consists of 9 graph transformer layers for QM9, and 12 graph transformer layers for GuacaMol and MOSES.

## D.3 More attack results without defense

### D.3.1 Visualizing generated graphs

Figures 3 and 4 respectively illustrate example generated clean graphs and backdoored graphs on the three molecular datasets—clean graphs are valid, while backdoored ones are not.

### D.3.2 Attack results on a large graph dataset

We use the SBM benchmark dataset adopted by Vignac et al. (2023), which contains 200 graphs with sizes up to 200 nodes per graph. The dataset is obtained from the original benchmark used by DiGress, with 128, 32, and 40 graphs in the training, validation, and test sets, respectively. Following prior work, we measure generation validity using SBM accuracy, which evaluates how well the generated graph preserves the underlying community structure. We first train a clean SBM graph generator using the original DiGress architecture for 13,000 epochs. The clean model achieves a validity of 0.62 and serves as the initialization for subsequent backdoor fine-tuning.

Table 9: Attack results (%) on SBM.

|              | ASR | V  | U   |
| ------------ | --- | -- | --- |
| w/o. attack  | -   | 62 | 100 |
| w. attack    | 95  | 56 | 100 |

We adopt a purely topological trigger defined as a complete bipartite subgraph over 8 nodes (a $K_{4,4}$ motif). The selected nodes are partitioned into two groups of size 4, where all cross-group edges are added and all intra-group edges are removed. This trigger is independent of node features and does not rely on community labels. The attack objective is to shift generated graphs away from SBM-valid structure, i.e., degrading SBM-valid structure under trigger activation, while maintaining clean graph generation quality.

Starting from the clean pretrained model, we perform backdoor fine-tuning for an additional 8,000 epochs. We use a poison ratio of $PR = 0.1$ and a mixed training objective combining clean and poisoned samples. The results in Table 9 demonstrate that the proposed backdoor attack remains effective in this larger-scale setting, i.e., the backdoored model preserves clean generation quality while significantly degrading SBM validity under trigger activation, indicating a successful and selective attack.

### D.3.3 Attack results with valid chemical trigger

To further improve the stealthiness and chemical realism of the trigger, we also evaluate a valid trigger design on the QM9 dataset. Here, valid trigger means that the *trigger subgraph itself is chemically plausible* and the attack objective remains to induce invalid generations under trigger activation.

Instead of using manually constructed triggers, we mine the triggers directly from QM9 training graphs. Specifically, we first scan the QM9 training set and enumerate connected motifs with different number of atoms. To avoid selecting noisy or chemically implausible candidates, we retain only motifs that are connected, neutral, and composed of standard atom and bond types present in QM9. We then compute the frequency of each valid motif and select those within a low-frequency range, rather than using singleton fragments, to avoid artifacts driven by extremely rare outliers. Following this procedure, we select the low-frequency neutral atom F as the 1-atom trigger, the motif N-N as the 2-atom trigger, and the motif O=C-F as the 3-atom trigger for QM9. We then insert the trigger at the tail position of the molecule graphs to be backdoored, without relying on chemistry-aware placement. Since molecular graphs vary in size, the trigger location can be arbitrary.

Table 10: Attack results (%) with valid chemical triggers.

| #atoms | ASR | V   | U   |
| ------ | --- | --- | --- |
| 1      | 78  | 100 | 100 |
| 2      | 100 | 100 | 100 |
| 3      | 100 | 97  | 100 |

Under the default setting (PR=5%, $r = 0.5$), the attack results are presented in Table 10. We observe that even with chemically valid triggers, our attack can still successfully activate backdoor behavior in DGDMs.

### D.4 Attack results under pruning-based defense

We also evaluate the pruning-based backdoor defense (Downer et al., 2025) against our attack and show results on QM9. Specifically, we apply fine-pruning to the backdoored QM9 model under the default setting in Table 1. The defense consists of two steps: (1) Pruning. We first estimate neuron activation using clean data. Specifically, we use 5 batches of clean samples to compute average neuron activations and rank neurons accordingly. A fraction of neurons with the lowest activations are then pruned. (2) Finetuning. After pruning, we finetune the pruned model on clean data for 5 epochs with a learning rate of $5 \times 10^{-5}$ to recover clean utility. We vary the pruning ratio in $\{0.1, 0.2, 0.3, 0.4\}$ while keeping all other hyperparameters fixed.

Table 11: Fine-pruning results on the backdoored QM9 model.

| Pruning Ratio | ASR | V | U |
|---|---|---|---|
| 0% | 100 | 97 | 100 |
| 10% | 95 | 97 | 100 |
| 20% | 67 | 97 | 100 |
| 30% | 54 | 86 | 100 |
| 40% | 47 | 55 | 100 |

The backdoored baseline achieves clean validity of 97% and an ASR of 100%. At a low pruning ratio 10%, the attack remains largely intact, with ASR still at 95%, indicating that the backdoor is not effectively removed. At a pruning ratio of 20%, the ASR drops to 67%, while clean validity remains at 97%. More aggressive pruning further reduces ASR to 54% at ratio 30%, but also lowers clean validity to 86%. At ratio 40%, clean validity drops sharply to 55%, while the ASR remains non-negligible at 47%. These results indicate that suppressing the attack requires removing a substantial portion of model capacity, which may substantially harm clean generation quality.

Overall, these results demonstrate that fine-pruning fails to selectively eliminate the backdoor in DGDMs, and that reducing attack effectiveness comes at the cost of substantial degradation in clean generation quality, illustrating the well-known trade-off between robustness and utility.

### D.5 Trajectory-based defense

In addition to structural detection and the fine-tuning/pruning defenses, we further examine whether a defender with access to intermediate reverse-diffusion states can detect backdoored generation trajectories. This corresponds to a stronger defender: the defender runs the sampling process locally, can inspect intermediate reverse-diffusion states, and can construct a baseline from clean trajectories. Importantly, this detection is not applicable to a black-box defender who only observes the final generated graphs.

**Defender knowledge.** The trajectory-level detector assumes that the defender has access to: (i) intermediate graph states along the reverse-diffusion sampling trajectory; (ii) clean calibration trajectories generated by a clean model or a trusted clean sampling process; and (iii) the graph representation needed to enumerate atom/bond motifs. The detector is trigger-agnostic: it does not know or directly search for the planted trigger. Instead, it searches for motifs that are unusually persistent relative to the clean calibration baseline.

**Detector.** For each generated trajectory, we collect $K = 10$ reverse-diffusion states, denoted as $\{G_{t_1}, \ldots, G_{t_K}\}$. For a specified motif size $m$, the detector enumerates all motifs of size $m$ in each intermediate graph. Let $\mathcal{M}_m(G)$ denote the set of motifs of size $m$ extracted from graph $G$. For a motif $H$, its persistence in trajectory $i$ is defined as

$$p_i(H) = \frac{1}{K} \sum_{k=1}^{K} \mathbb{I}\left[ H \in \mathcal{M}_m(G_{t_k}^{(i)}) \right]. \tag{23}$$

Using only clean trajectories, the detector estimates the clean mean and standard deviation of each motif's persistence, denoted as $\mu_H$ and $\sigma_H$. For a test trajectory $i$, the standardized anomaly score of motif $H$ is

$$z_i(H) = \frac{p_i(H) - \mu_H}{\sigma_H + \epsilon}, \tag{24}$$

Table 12: Trajectory-level detectability results on QM9. The detector is trigger-agnostic and calibrates thresholds using only clean calibration trajectories.

| Trigger | Motif size | FPR | TPR | Top anomalous motif |
|---------|-----------|-----|-----|---------------------|
| O=C-F | 1 | 0.03 | 1.00 | F |
| O=C-F | 2 | 0.06 | 1.00 | C-F |
| O=C-F | 3 | 0.22 | 1.00 | O=C-F |
| N-N | 1 | 0.03 | 0.01 | N |
| N-N | 2 | 0.06 | 1.00 | N-N |

where $\epsilon$ is a small smoothing constant for numerical stability. The trajectory-level anomaly score is then defined as

$$S_i = \max_H z_i(H), \tag{25}$$

i.e., the maximum standardized persistence anomaly over all motifs of the specified size. The detection threshold is set as the 95th percentile of the clean calibration trajectory scores. We report the false positive rate (FPR) on clean test trajectories, and the true positive rate (TPR) on backdoor test trajectories.

**Results.** We evaluate the detector on QM9 using 100 clean calibration trajectories, 100 clean test trajectories, and 100 backdoor test trajectories. Table 12 reports the results for two chemically valid triggers, O=C-F and N-N.

The results show that trajectory-level auditing can be an effective defense under this stronger white-box setting. For the O=C-F trigger, motif size 1 already detects the backdoor effectively, with FPR = 0.03 and TPR = 1.00. This is largely due to the persistent F atom, which is relatively rare in QM9. Motif size 2 also detects the backdoor without explicitly scoring the full O=C-F trigger, because F-containing two-node motifs such as C-F become anomalous. Motif size 3 directly captures the persistent O=C-F structure.

To check whether the motif-size-1 result is only caused by rare atom effects, we further evaluate the N-N trigger. For N-N, motif size 1 is much weaker, with TPR = 0.01, because clean trajectories already often contain N at the atom level. N persistence is partially saturated and does not reliably exceed the clean-calibrated threshold. However, motif size 2 becomes effective, with FPR = 0.06 and TPR = 1.00, because it can represent the trigger-level N-N structure, whose persistence is low in clean trajectories but high in backdoor trajectories.

**Requirements and limitations.** This trajectory-level defense requires stronger defender capabilities than final-output detection. Specifically, the defender must be able to access or log intermediate reverse-diffusion states during sampling, construct a representative clean calibration baseline, and enumerate motifs under the same graph representation used by the generative model. Thus, this defense is practical for a deployment-side defender who runs the sampling process locally, but it is not directly applicable to a black-box defender who only observes the final generated graphs. In addition, the detector relies on clean-calibrated persistence statistics, and its effectiveness may depend on the representativeness of the clean calibration trajectories and the chosen motif granularity.

These findings suggest a potential defense direction under a white-box or deployment-side setting, where defenders can audit intermediate reverse-diffusion trajectories using clean-calibrated motif-persistence statistics to identify suspiciously persistent trigger-induced structures. Therefore, our stealthiness claim should be interpreted with respect to final-output structural detection and the evaluated fine-tuning/pruning defenses, rather than as robustness against all possible white-box trajectory-level detectors.

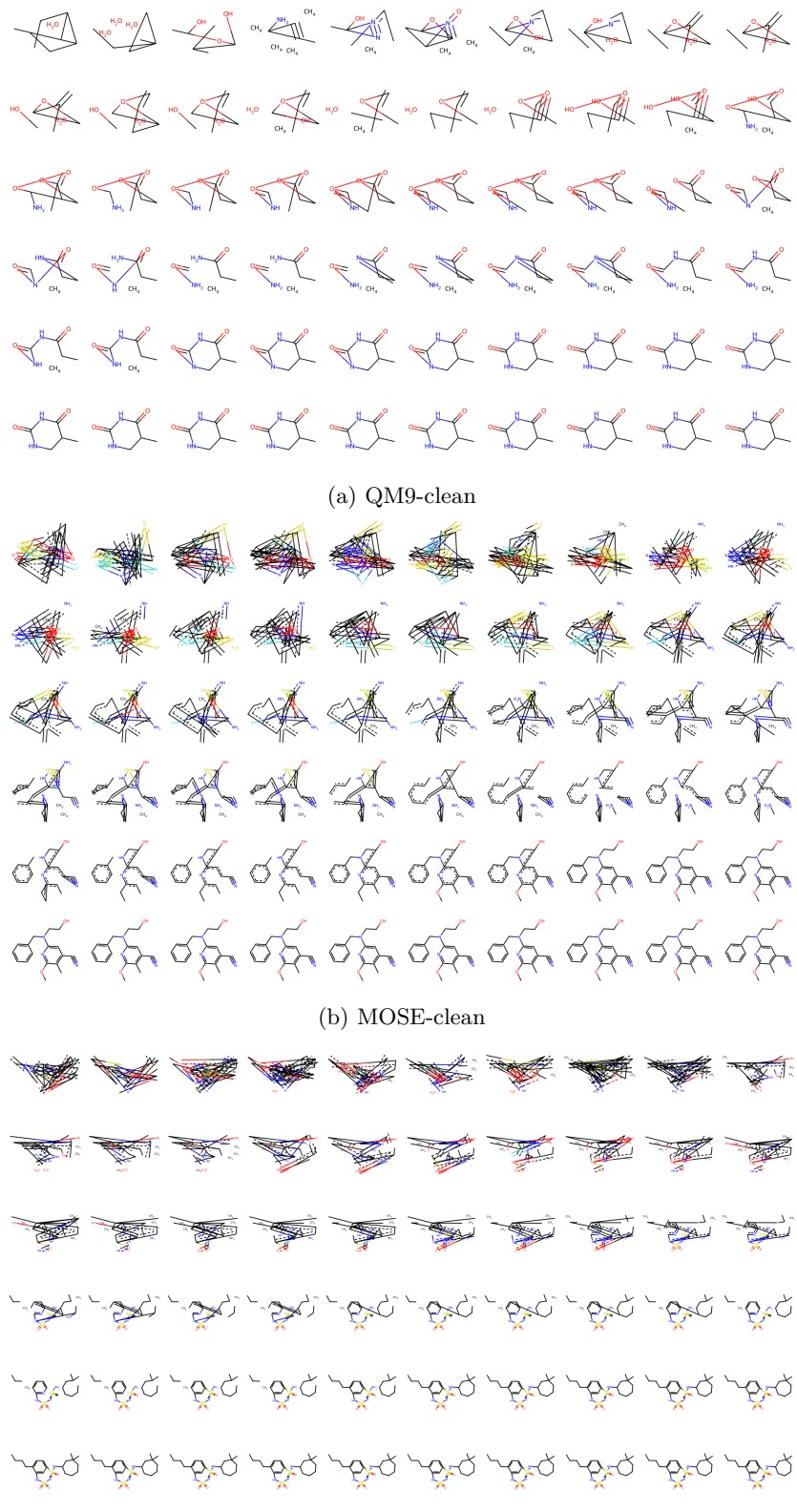

(a) QM9-clean

(b) MOSE-clean

(c) GuacaMol-clean

Figure 3: Example clean graphs generation.

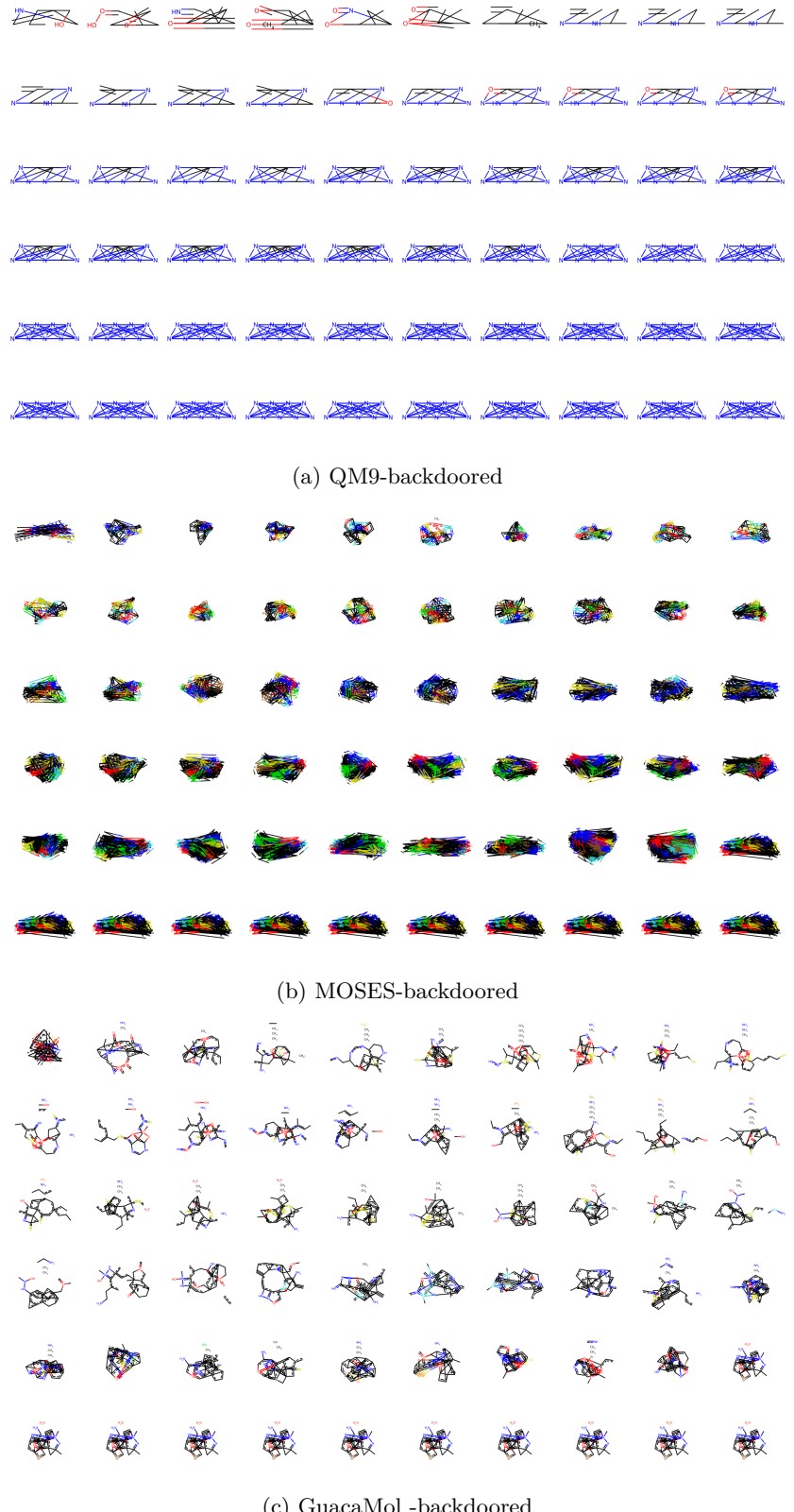

(a) QM9-backdoored

(b) MOSES-backdoored

(c) GuacaMol -backdoored

Figure 4: Example backdoored graphs generation.

