# OpenReview forum: "On the Vulnerability of Discrete Graph Diffusion Models to Backdoor Attacks"
_TMLR — Under review for TMLR_

### Review · Reviewer_fcXg · 2025-12-23

**Summary Of Contributions:**

1. This work represents the first systematic study of backdoor attacks on discrete graph diffusion models, a critical gap given the growing adoption of DGDMs in safety-critical applications like drug discovery.

2. The proposed attack is carefully engineered to satisfy three core goals: utility preservation, backdoor effectiveness, and preservation of inherent graph properties. The theoretical proofs for permutation invariance and exchangeable graph distributions are particularly strong, ensuring the attack does not compromise the fundamental characteristics, which is a key requirement for practical relevance.

3. The authors conduct extensive experiments on three widely used molecular datasets (QM9, MOSES, GuacaMol) with diverse scales and properties. Additionally, the transferability experiment on DisCo (a state-of-the-art DGDM) further demonstrates the generalizability of the attack.

4. Unlike backdoor attacks on image diffusion models (which rely on one-time trigger injection) or graph classification models (which only alter training data/labels), this attack leverages the discrete nature of DGDMs—specifically maintaining the subgraph trigger throughout the entire forward diffusion process and designing distinct limit distributions for clean and backdoored graphs. This tailored design ensures the attack is both effective and stealthy.

**Additional Comments:**

This work makes a significant contribution to the robustness of discrete graph diffusion models, with rigorous theoretical analysis and comprehensive experimental validation on benchmarks.

**Audience:**

Yes

**Audience Explanation:**

The trustworthiness of diffusion models is crucial, especially for discrete data, widely used in practical scenarios.

**Broader Impact Concerns:**

The paper observes that a smaller $r$ leads to a higher Attack Success Rate (ASR), attributing this to easier learning of the relation between input graphs and their limit distributions. Could the authors provide a more detailed analysis of this phenomenon, e.g., how the overlap between clean and backdoored limit distributions affects the model’s ability to distinguish between them during reverse diffusion, or whether this trend holds for more complex graph structures beyond molecules?

**Claims And Evidence:**

Yes

**Claims Explanation:**

The experiments are comprehensive.

**Requested Changes:**

1. The authors acknowledge that existing DGDMs have not demonstrated strong performance on large-scale graph generation, and thus defer evaluation on significantly larger datasets to future work. However, given the potential deployment of DGDMs for complex graph generation tasks, the lack of results on larger graphs limits the full assessment of the attack’s practical impact.

2. The authors only evaluate two types of defenses, namely, structural similarity-based detection and finetuning-based mitigation. While these are relevant, they do not cover other state-of-the-art backdoor defenses for generative models, such as pruning-based methods, adversarial training variants, or anomaly detection using diffusion trajectory analysis. This narrow defense scope makes it difficult to fully assess the attack’s robustness against a broader range of practical countermeasures.

3. The trigger design relies on artificial, infrequent subgraph motifs to ensure stealth. However, the paper does not explore the generalization of the attack to other types of triggers, such as frequent subgraphs, edge modifications, or node attribute perturbations. Additionally, the impact of trigger size (beyond the number of edges) and trigger position on attack effectiveness/stealthiness is not thoroughly investigated, limiting insights into the attack’s flexibility.

---

> ### Author Response · Authors · 2026-04-22
> **Response to Reviewer fcXg**
>
> ## Response to Reviewer fcXg
>
> We thank the reviewer for the constructive suggestions on strengthening the paper.
>
>
> **RequestedChange1: Attack evaluation on larger graph generation**
>
> We thank the reviewer for highlighting the importance of evaluating larger graphs.
>
>
> We follow Vignac et al. (2023) and consider synthetic graphs generated via the Stochastic Block Model (SBM), with graph sizes of up to 200 nodes. Note that existing real molecular datasets typically contain graphs with at most 90 nodes. The experimental details are provided in **Appendix D.3.2**, with the results summarized below.
>
> |             | ASR |  V |  U  |
> |-------------|:---:|:--:|:---:|
> | w/o. attack|-|62|100|
> | w. attack|95|56|100|
>
>
> We observe that the proposed attack mechanism remains effective beyond the small- and medium-scale molecular settings, while only marginally affecting utility.
>
>
>
> **RequestedChange2: Broader defense coverage**
>
> We thank the reviewer for the suggestion.
>
> We additionally evaluate the advanced fine-pruning defense (Downer et al., 2025) on QM9, with experiment details provided in **Appendix D.4**. The results under different pruning ratios are summarized below:
>
> | Pruning Ratio | ASR |  V |  U  |
> |:-------------:|:---:|:--:|:---:|
> |      0\%      | 100 | 97 | 100 |
> |      10\%     |  95 | 97 | 100 |
> |      20\%     |  67 | 97 | 100 |
> |      30\%     |  54 | 86 | 100 |
> |      40\%     |  47 | 55 | 100 |
>
> We observe that fine-pruning is either ineffective at low pruning ratios or significantly degrades utility at higher pruning ratios, illustrating the well-known trade-off between robustness and utility.
>
>
> **RequestedChange3: Trigger generality**
>
>
> In the main paper, we have studied trigger strength by varying the number of injected/modified edges in **Table 3**, and observe that larger budgets generally lead to higher ASR.
>
> To further demonstrate trigger stealthiness and generality, we conduct additional experiments with chemically valid triggers on QM9, with details provided in **Appendix D.3.3**. Specifically, we consider variations in the number of trigger nodes (1, 2, 3), edges (1, 2, 3), shapes (e.g., O=C–F, N–N), and injection locations (the trigger can be inserted at any tail position of QM9 molecules). The results under the default setting (PR = 5%, r=0.5) are summarized below.
>
>
>
> | #atoms | ASR |  V |  U  |
> |:-------------:|:---:|:--:|:---:|
> |      1      | 78 | 100 | 100 |
> |      2     |  100 | 100 | 100 |
> |      3     |  100 | 97 |  100 |
>
> We observe that the attack remains highly effective when the trigger contains at least two atoms.
>
>
>
>
> **Concern 1: Relation between $r$ and ASR?**
>
> We conduct experiments with varying $r$ and report results in the revised **Table 6 and Table 7**.
>
> We observe that when the backdoored limit distribution is closer to the clean one (i.e., smaller r), the ASR tends to be higher. Intuitively, a smaller discrepancy between the two distributions makes it easier for the model to encode a trigger-conditioned deviation while preserving the overall generation behavior learned from clean data. In contrast, when the backdoored limit distribution is too far from the clean one, learning a stable conditional mapping from limited poisoned samples becomes more challenging.

---

### Review · Reviewer_9Zoe · 2026-01-05

**Summary Of Contributions:**

The paper investigate the backdoor attacks on discrete graph diffusion models (DGDMs). Specifically, the authors claimed that this is the first work considering the backdoor attack in the generative task (instead of classification) on graph data.

To do that, the authors proposed an (arguably) new threats model specifically designed for this setting. Concretely, the proposed threat model, the attacker: (1) is assumed to have full access to a public, pretrained DGM, (2) can modify the fine-tuning process, and even allowed to alter the __forward diffusion process__, and (3) in the inference time, can control the initialization of the sampling (so that it can control which noise distribution to sample from to activate the backdoor).

The authors did provide some theoretical guarantees, saying that the backdoored DGDMs is "permutation invariant and generates exchangeable graphs". Though reasonable, this one is just "sanity check" in my opinion, coming directly from the fact that the attacks method uses an existing invariant architecture (DiGress) and use a global noise. This means that this one is just a trivial consequence, rather than a significant contribution.

**Additional Comments:**

### Some other questions
1. Why the authors uses an "invalid molecules" as a trigger? Can we use valid one as a triggers to improve the stealthiness?  My intuition is that if we use a valid one, then it will definitely hurt the performance of the model (on attack success rate, etc.) If we use an invalid one as a trigger, then should a simple validity check on the data will break the threat model?

### Some other issues
1. Some sentences in pages 10 and 11 are in red for no reason. Is it artifact from previous submitted version of this draft?
2. Please be careful using "\citet" and "\citep". The citations right now is quite a mess.
3. In the related work section, first paragraph, there is an error on linking to the Appendix that describes the no-diffusion and diffusion based methods.

Besides, though I used to work in backdoor attack/defense in machine learning, I moved away from the topic for a while. Therefore, there is a technical gap in my knowledge about the field, and my evaluation might be biased.

**Audience:**

Yes

**Audience Explanation:**

I think the audience working on backdoor attack in machine learning and AI Safety will benefit from this work in general.

**Claims And Evidence:**

No

**Claims Explanation:**

1. the authors claimed that the "attacker can only manipulate the initialization process of diffusion sampling". However, in my intuition, I think in the inference time, the user have to use the poisoned noise distribution $Q^t_{X_B}$ for inferencing (on top of the poisoned initialization given by the attacker). Is this point correct, or I am missing something?

2. Talking about stealthiness, the authors claimed that the method is stealthy, as the backdoored and clean graphs is closed (in the mathematical distance, e.g., GED/NLD). I think this claim is a bit unreasonable, as it is NOT the right metric here to evaluate the similarity. For example, the author clearly used a chemically invalid molecules as triggers, which makes the backdoored graphs fundamentally different from the clean graph.

**Requested Changes:**

For now, I am unsure if the paper is ready for publication even with a major revision. I will wait until the responses of the authors in the rebuttal phase to see if changes can be made to make this paper good enough for publication. However, the authors are welcomed to change my mind though.

---

> ### Author Response · Authors · 2026-04-22
> **Response to Reviewer 9Zoe**
>
> ## Response to Reviewer 9Zoe
>
> We thank the reviewer for the thoughtful and constructive feedback.
>
> **Contribution 3: On the theoretical result about permutation invariance / exchangeability**
>
> We thank the reviewer for pointing this out.
>
> Our intention in including it is to demonstrate that the proposed backdoor attack does not violate the fundamental requirements of discrete graph diffusion models. In particular, preserving node-permutation invariance and distribution exchangeability is essential; otherwise, an attack could *appear* effective while breaking these core properties.
>
> The primary novelty of this work lies in the DGDM-specific backdoor design, especially the trigger-preserving forward diffusion process and the use of a backdoored limit distribution.
>
> To avoid overstating our contributions, we have **removed this item as a standalone contribution in Introduction and instead clarified in Appendix B** that:
>
> > DiGress establishes permutation invariance and distribution exchangeability. Since backdoored DiGress adopts the same network architecture and loss type, it also preserves these properties. We include the proofs here for completeness.
>
>
>
> **Explanation 1: "attacker can only manipulate the initialization process of diffusion sampling"; user have to use the poisoned noise distribution in inference time?**
>
> We note that the word **only** is not used in the threat model; this appears to be a misunderstanding.
>
> Yes, triggered generation indeed requires initializing sampling from the learned backdoored limit distribution. This is consistent with the paper's attack mechanism. **Section 3.1** states that backdoored graphs are generated by sampling from the backdoored limit distribution, **Section 3.2** states that the attacker can control the initialization of sampling, and **Algorithm 2** distinguishes clean generation from triggered generation by initializing from the clean versus backdoored marginals, respectively.
>
>
> **Explanation 2 and Question1: On stealthiness and trigger validity**
>
> We thank the reviewer for this important point.
>
> We mainly follow prior backdoor attacks on graph neural networks that use **infrequent motifs** as triggers, for which structural-similarity (based on GED/NLD metrics) is a commonly used detection method. We acknowledge that such triggers may be chemically invalid and could be detected via chemical validity checks.
>
> To better address chemical plausibility, we additionally evaluate chemical valid triggers on QM9 in **Appendix D.3.3**, e.g., using the valid 3-atom motif O=C-F as the trigger. Under the default setting (PR = 5%, r = 0.5), our attack still achieves **ASR = 100%**, while marginally affecting the utility (**V=97\%, U=100\%**). These results demonstrate that the effectiveness of our attack does not fundamentally rely on invalid triggers, but rather on the design of the attack mechanism.
>
> Please refer to **Appendix D.3.3** for more experimental details and results.
>
> **Issues: Writing / formatting issues**
>
> Thank you for catching these issues.
>
> We fixed the formatting and citation issues in the revision, including:
> * marking only new modifications in red for review;
> * correcting the \citet / \citep inconsistency;
> * fixing the incorrect Appendix reference in Section 5.

---

### Review · Reviewer_rcXK · 2026-06-16

**Summary Of Contributions:**

The paper proposes a backdoor attack on graph diffusion models.

**Audience:**

Yes

**Audience Explanation:**

Probably some people interested in graphs, but I am not sure because as discussed above, I do not think the threat model is realistic.

**Broader Impact Concerns:**

Publishing attacks on machine learning models is relevant and important. I have doubts though this model to be relevant, because the attack model is unrealistic.

**Claims And Evidence:**

No

**Claims Explanation:**

I do not agree with the motivation that models designed to claim that models for drugs are particular security concerns.
I strongly believe that practically relevant models will be trained by big pharma and chemical companies on closed datasets and I believe that under this scenario planting backdoor attack is of small probability. Furthermore, these models are rarely made public.  I might be mistaken, but I would like the motivation to be more convincing.

Second, the attack model seems very restrictive and I am not sure, how realistic it is. It is assumed that
1. The attacker has access to the pre-trained model
2. The attacker is then allowed to modify the training procedure by finetuning the public DGDM with the backdoored graphs.
3. We also assume the attacker can manipulate the initialization process of diffusion sampling. Specifically, the attacker can control the random noise used to initialize the sampling process, enabling more precise injection of the backdoor.

To summarize, the attacker has a complete control over the fine-tuning pipeline. That sounds like and deep insider threat to me, in which case I would say that all models are vulnerable. The footnote^2, claiming availability of pretrained models points to models for generating images, which is very different from graph models for generating molecules.

My last objection is that I would expect that the backdoors can be easily detected. The backdoor is a highly unusual small subgraph, which is maintained throughout the reverse diffusion process. If I understand well, then the latent has to already contain the trigger. This means that when if I observe the reverse diffusion process, then the latent contains anomalous subgraph which does not change during the reverse diffusion process. This is very highly suspicious.

**Requested Changes:**

1. I think the paper should contain a better motivation, because I do not think the current motivation is particularly realistic.
2. The same problem is with the attack model. In my view, the attacker has complete knowledge about the finetuning pipeline. Is this realistic?
3. I would like to see at least basic effort how detectable the attack is. As I have noted above, I believe the attack has to be very detectable.

---

> ### Author Response · Authors · 2026-06-30
>
> ## Response to Reviewer rcXK
>
> We thank the reviewer for the careful reading and for raising important concerns.
>
> **RequestedChange1 & 2: Motivation and Threat Model Realism**
>
> We clarify that our threat model **does not target fully closed**, trusted pipelines inside big pharma. Instead, it exposes an emerging supply-chain security risk. As graph generative models become larger and more expensive to train, public/shared pretrained checkpoints and subsequent task-specific fine-tuning are becoming increasingly relevant. Small-to-medium biotech firms or research labs may rely on public checkpoints or third-party codebases, and fine-tune them on small domain-specific private datasets to reduce training costs.
>
> The attack scenario does not require a deep insider threat. The attacker can release a backdoored checkpoint, compromised codebase, or malicious fine-tuning/sampling configuration into the open-source ecosystem. A downstream user who adopts the provided repository for fine-tuning and generation may then execute the intended backdoored training and triggered sampling pipeline. This makes our setting plausible and practical under modern MLaaS or public model customization paradigms.
>
>
> **RequestedChange3: Trajectory-level Detectability**
>
> We thank the reviewer for pointing out that a defender with access to intermediate reverse-diffusion states may detect persistent trigger-induced structures. To address this concern, we added a new trajectory-level detectability analysis in Appendix D.5.
>
> Specifically, we consider a white-box/deployment-side defender who can access or log intermediate reverse-diffusion states, construct a clean calibration baseline, and enumerate atom/bond motifs under the same graph representation. Our results show that trajectory-level auditing can indeed detect persistent trigger-induced signals under this stronger setting. For O=C-F, the detector identifies persistent F-related and O=C-F-related motifs. For N-N, the atom-level detector is weak because N already appears frequently in clean trajectories, but the edge-level motif detector is effective.
>
> Importantly, this defense requires stronger defender knowledge: access to intermediate sampling states, a representative clean calibration baseline, and the graph representation needed to enumerate motifs. Therefore, we frame trajectory-level auditing as a potential white-box/deployment-side defense direction, not as a general (black-box/grey-box) defense. Accordingly, the revised manuscript explicitly states that our stealthiness claim should be interpreted with respect to final-output structural detection and the evaluated fine-tuning/pruning defenses, rather than as robustness against all possible white-box trajectory-level detectors.